



# 1 Consistent assimilation of multiple data streams in a
# 2 carbon cycle data assimilation system

**Natasha MacBean[1], Philippe Peylin[1], Frédéric Chevallier[1], Marko Scholze[2],**
**Gregor Schürmann[3]**
[1]{Laboratoire des Sciences du Climat et de l'Environnement, LSCE/IPSL, CEA-CNRS-
UVSQ, Université Paris-Saclay, F-91191 Gif-sur-Yvette, France}
[2]{Department of Physical Geography and Ecosystem Science, Lund University, Lund,
Sweden}
[3]{Max Planck Institute for Biogeochemistry, Jena, Germany}
Correspondence to: N. MacBean (nlmacbean@gmail.com)

### 13 Abstract

Data assimilation methods provide a rigorous statistical framework for constraining
the parametric uncertainty of land surface models (LSMs), with the aim of improving our
predictive capability as well as identifying areas in which the models need improvement. The
increase in the number of available datasets in recent years allows us to address different
aspects of the model at a variety of spatial and temporal scales. However, combining data
streams in a DA system is not a trivial task. In this study we highlight some of the challenges
surrounding multiple data stream assimilation, with a particular focus on the carbon cycle
component of LSMs. We examine the impact of biases and inconsistencies between the
observations and the model (resulting in non Gaussian error distributions) and the impact of
non-linearity in model dynamics. In addition we explore the differences between performing a
simultaneous assimilation (in which all data streams are included in one optimisation) and a
step-wise approach (in which each data steam is assimilated sequentially), given the
assumptions inherent to the inversion algorithm chosen for this study. We demonstrate some
of these issues by assimilating synthetic observations into two simple models: the first a
simplified version of the carbon cycle processes represented in many LSMs, and the second a
non-linear toy model. We further discuss these experimental results in the context of recent





studies in the carbon cycle data assimilation literature, and finally we provide some
perspectives and advice to other land surface modellers wishing to use multiple data streams
to constrain their models.
Keywords: data assimilation, carbon cycle, biogeochemical cycles, land surface model
**1   Introduction**
The carbon cycle is an important component of the Earth system, especially when
considering the climatic impact of rising greenhouse gases concentrations from fossil fuel
emissions and land use change. It is estimated that the oceans and land surface absorb
approximately half of the $CO_2$ emissions due to anthropogenic activity, but uncertainties
remain in the strength and location of sources and sinks, as well as in predictions of future
trends (Ciais et al., 2013). Observations allow us to understand the system up until the present
day, but they cannot tell us about the future, and can be limited in their spatial coverage. They
also cannot distinguish between the complex interactions that may occur between different
processes. Incorporating our current knowledge of physical mechanisms of biogeochemical
cycles, including carbon, C, dynamics, into Land Surface Models (LSMs) represents a
promising approach to analyse these interacting effects, to upscale observations to larger
regions, and to make future predictions. However, the models can be limited by the lack of
process representation, either due to gaps in our knowledge or in our technical and computing
capability. As a result, model evaluations reveal that not all variables are well-captured by the
model under current conditions (Anav et al., 2013), and the spread between model projections
is still very large (Sitch et al., 2015).
Aside from model structural and forcing errors, one source of uncertainty is related to the
parameter (i.e. fixed) values of a model. Model-data fusion, or data assimilation (DA), allows
the calibration, or optimisation, of these values by reducing the model-data misfit while
accounting for the uncertainties inherent in both the model and data in a statistically rigorous
framework. The C cycle component of most LSMs is complex and contains a large number of
parameters. Luckily however, there are an increasing number of in-situ and remote sensing-
based data streams that can be used for parameter optimisation. These data bring information
on different spatial and temporal scales, such as:



• Atmospheric $CO_2$ concentration data measured at surface stations at continental to
global scales, which provide information from synoptic timescales to inter-annual
variability (IAV) and long-term trends.
• Eddy covariance net $CO_2$ (net ecosystem exchange – NEE) and latent (LE) and
sensible heat fluxes measured at half-hourly intervals at many sites across different
ecosystems/regions, providing information at seasonal to inter-annual timescales.
• Satellite-derived measures of vegetation dynamics, including "greenness" indices (i.e.
the Normalised Difference Vegetation Index – NDVI), fraction of absorbed
photosynthetically active radiation (FAPAR) and leaf area index (LAI) at global scales
and at daily time step spanning more than a decade, thus capturing IAV and long-term
trends (though usually with a trade-off between spatial and temporal resolution).
• Satellite-derived measurements of soil moisture and land surface temperature at the
same temporal and spatial scales as the satellite-derived observations of vegetation
productivity.
• Aboveground biomass measurements are currently taken at only one or a few points in
time at plot scale up to regional scale from aircraft and satellite data, or are estimated
from allometric relationships at each site.
• Soil C stock estimates usually are only taken at one point in time at plot scale.
• Ancillary data on vegetation characteristics such as tree height or budburst – one
measured at certain well-instrumented sites.
Increasingly, researchers are attempting to bring these sources of information together to
constrain different parts of a model at different spatio-temporal scales within a multiple data
stream assimilation framework (e.g. Richardson et al., 2010; Keenan et al., 2012; Kaminski et
al., 2012; Forkel et al., 2014; Bacour et al., 2015). However, whilst the potential benefit of
adding in extra data streams to constrain the C cycle of LSMs is clear, multiple data stream
assimilation is not as simple as it may seem. When using more than one data stream there is
the option to include all data streams together in the same optimisation (simultaneous
approach), or to take a sequential (step-wise) approach. Mathematically, the optimal approach
is the simultaneous, but computational constraints related to the inversion of large matrixes or



the requirement of numerous simulations (especially for global datasets), and/or the weight of different data streams in the optimisation, may complicate a simultaneous optimisation. On the other hand, in a step-wise assimilation the parameter error covariance matrix has to be propagated at each step, which implies that it can be computed. If the parameter error covariance matrix can be properly estimated and is propagated between each step, the step-wise approach can be mathematically equal to simultaneous. However, many inversion algorithms (e.g. derivative based methods that use the gradient of the cost function to find its minimum) require assumptions of model (quasi-) linearity and Gaussian parameter and observation error distributions. If these assumptions are violated, or the error distributions are poorly defined, it is likely that the step-wise will not be equal to the simultaneous, and that information will be lost at each step. An incorrect description of the observation (– model) error distribution could result from the wrong assumption about the distribution of the residuals between the observation and the model, a poor characterisation of the error correlations, an incompatibility between the model and the data (possibly due to a model structural issue or differences in how a variable is characterised), or a bias in the observations that is not unaccounted for (i.e. is treated as a random error). Whilst a simultaneous optimisation is mathematically more rigorous in the sense that the error correlations are treated within the same inversion, if the prior distributions are not properly characterised any bias may be aliased to the wrong parameters (Wutzler and Carvalhais, 2014), more so than in a step-wise approach.

This tutorial-style paper demonstrates some of the challenges of multiple data stream assimilation discussed above with two simple models: one a simplified version of the carbon dynamics included in many LSMs, and the other a "toy" model designed to demonstrate the issues that arise with complex, non-linear models. Section 2 provides a description of these models, the inversion algorithm used to optimise the model parameters and the experiments performed, followed by the results for each test case. Section 3 further discusses the challenges outlined in Section 2 with reference to recent carbon cycle multiple data stream assimilation studies in the literature. Finally Section 4 provides some advice to land surface modellers wishing to carry out multiple data stream assimilation.



**2    Demonstration with two simple models and synthetic data**
**2.1    Methods**
### 2.1.1  Simple carbon model
To demonstrate the challenges of multiple data stream assimilation in a carbon cycle
context, we have chosen a test model that represents a simplified version of the carbon cycle
dynamics typically implemented in most LSMs. The model has been well-documented in
Raupach (2007) and has been used previously in the OptIC DA inter-comparison project
(Trudinger et al., 2007). It is based on two equations that describe the temporal evolution of
two carbon pools, $s_1$ and $s_2$:

$$\frac{ds_1}{dt} = F(t)\left(\frac{s_1}{p_1 + s_1}\right)\left(\frac{s_2}{p_2 + s_2}\right) - k_1 s_1 + s_0 \tag{1}$$

$$\frac{ds_2}{dt} = k_1 s_1 - k_2 s_2 \tag{2}$$

In this model formulation, $s_1$ and $s_2$ are approximately equivalent to above- and belowground
biomass stocks. The unknown parameters $p_1$, $p_2$, $k_1$ and $k_2$ will be optimised in the inversions.
The first term on the right-hand side of Eq. (1) corresponds to the Net Primary Production
(NPP) i.e. the carbon assimilated into the system as a function of time, $F(t)$, weighted by
factors that account for the size of both pools in order to introduce a limitation on NPP (the
two fractions in parentheses). The litterfall is an output of $s_1$ and an input to $s_2$ and is a
constant fraction of the aboveground carbon reserve as represented by $k_1 s_1$. Heterotrophic
respiration (Rh) is an output of $s_2$ and is represented $k_2 s_2$. The constant $s_0$ is a "seed
production" term set to 0.01 (i.e. not optimised) to ensure the model does not verge towards
zero. A more detailed description of the properties of the model is given in Trudinger et al.
(2007) and an in-depth analysis of the model behaviour is provided in Raupach (2007).
Synthetic observations of both $s_1$ and $s_2$ variables were used to optimise all the unknown
parameters in the model (see Section 2.1.5).
### 2.1.2  Non-linear toy model
Although the simple carbon model contains a non-linear term it is essentially still a
quasi-linear model. In order to illustrate the challenges associated with multiple data stream



data assimilation for more complex non-linear models, we defined a simple non-linear toy
model based on two equations with two unknown parameters:

$$s_1 = a\exp^b + at^2 \tag{3}$$

$$s_2 = \sin(10a + 10b) + 10t^2 \tag{4}$$

where $s_1$ and $s_2$ also correspond to two model state variables (as for the simple C model), $a$
and $b$ are the unknown parameters included in the optimisation, and $t$ is the independent
variable, which could represent time in a real-world scenario. Note that this model is not
based on any particular physical process associated with land surface biogeochemical cycles,
but it does contain typical mathematical functions that are observed in reality and
implemented in LSMs. For example, the sinusoidal function (Eq. (4)) could represent diurnal
variations of various processes such as photosynthesis and respiration. Exponential response
functions (such as in Eq. (3)) are also observed for certain processes, including the
temperature sensitivity of soil microbial decomposition. As for the simple carbon model,
synthetic observations corresponding to the $s_1$ and $s_2$ variables were used to optimise both
parameters (see Section 2.1.5).

### 2.1.3 Bayesian inversion algorithm

Most data assimilation approaches follow a Bayesian formalism which, simply put,
allows prior knowledge of a system (in this case the model parameters) to be updated, or
optimised, based on new information (from the observations). In order to achieve this we
define a "cost function" that describes the misfit between the data and the model, taking into
account their respective uncertainties, as well as the uncertainty on the prior information. If
we follow a Bayesian formalism and least-squares minimisation approach, and assume
Gaussian probability distributions for the model parameter and observation error
variance/covariance, we derive the following cost-function (Tarantola, 1987):

$$J(\mathbf{x}) = \frac{1}{2}[(H(\mathbf{x}) - \mathbf{y})^T . \mathbf{R}^{-1} . (H(\mathbf{x}) - \mathbf{y}) + (\mathbf{x} - \mathbf{x}^b)^T . \mathbf{B}^{-1} (\mathbf{x} - \mathbf{x}^b)] \tag{5}$$

where $\mathbf{y}$ is the observation vector, $H(\mathbf{x})$ the model outputs given parameter vector $\mathbf{x}$, $\mathbf{R}$ the
observation error covariance matrix (including measurement and model errors), $\mathbf{x}^b$ the a priori



parameter values, and **B** the prior parameter error covariance matrix. This framework leads to
a Gaussian posterior parameter probability distribution function.

3    The aim of the inversion algorithm is to find the minimum of this cost function,

thereby achieving the best possible fit between the model simulations and the measurements,
conditioned on their respective uncertainties and prior information. For cases where there is a
strong linear dependence of the model to the parameters (at least for variations in **x** of the size
of those expected in the data assimilation system), and where the dimensions of the problem
are not too large, the solution can be derived analytically. If not, as is usually the case with
LSMs, there are different numerical methods to find the most optimal parameter values.
These include global search methods that randomly search the parameter space and test the
likelihood of a particular parameter set at each iteration, and derivative methods, which
calculate the gradient of the cost function at each iteration to find its minimum. In this study
we use the latter class of methods. More specifically we use a quasi-Newton algorithm that
uses both the gradient of the cost function and its derivative (Hessian) to evaluate if the
minimum has been reached (i.e. where the gradient is zero). Thus we obtain the following
algorithm for iteratively finding the minimum (Tarantola, 1987, p195):

$$\mathbf{x}_{i+1} = \mathbf{x}_i + \varepsilon_i [\mathbf{H}^T \mathbf{R}^{-1} \mathbf{H} + \mathbf{B}^{-1}]^{-1} \mathbf{H}^T \mathbf{R}^{-1} (\mathbf{y} - H(\mathbf{x}) - \mathbf{B}^{-1} (\mathbf{x}_i - \mathbf{x}^b)) \tag{6}$$

where $i$ is the iteration number and **H** is the Jacobian, or first-order derivatives, of $H$, which in
this study is determined using a finite difference method. Note that as we are potentially
dealing with non-linear models, the quasi-Newton method has been slightly adapted to
include the constant scaling factor $\varepsilon_i$ (with a value <1.0) to ensure that the algorithm will
converge.

23    Of course no inversion algorithm is perfect, and therefore it is possible that the true

"global" minimum of the cost function has not been found. Derivative methods in particular
can get stuck in so-called "local minima", preventing the algorithm from finding the true
minimum. To address this issue we carry out a number of assimilations with different random
first guess points in the parameter space. If they all result in the same reduction in cost
function value, we can have more confidence that the true minimum has been found.

29    Once the minimum of the cost function has been found, the posterior parameter error

covariance can be approximated (using the linearity assumption) from the inverse Hessian of





the cost function around its minimum, which is calculated using the Jacobian of the model at
the minimum of $J(\mathbf{x})$ (for the set of optimized parameters), $\mathbf{H}_\infty$, following Tarantola (1987):
$$\mathbf{A} = [\mathbf{H}_\infty^T \mathbf{R}^{-1} \mathbf{H}_\infty + \mathbf{B}^{-1}]^{-1} \qquad (7)$$
Note that the posterior error covariance matrix can be propagated into the model space to
determine the posterior uncertainty on the simulated state variables as a result of the
parametric uncertainty (as shown in the coloured error bands in the time series plots – Figures
1 and 5) using the following matrix product and the hypothesis of local linearity (Tarantola,

8   1987):

$$\mathbf{R}_{post} = \mathbf{H}_\infty . \mathbf{A} . \mathbf{H}_\infty^T \qquad (8)$$
However, we do not detail the propagated posterior uncertainty on the state variables further
in this study; rather, we describe the impact of the optimisation on the model–data fit in terms
of the RMSE value and also in terms of the relative uncertainty reduction on the parameters.
## 2.1.4 Step-wise versus simultaneous assimilation
*Step-wise approach*
In the step-wise approach each data stream (in our cases $s_1$ and $s_2$, see above) is
assimilated sequentially, and the posterior error covariance matrix of Eq. (7) is propagated to
the next step as the prior in Eq. (6). Note that the error covariance matrix can only be
propagated if it is calculated within the inversion algorithm, which is the case here but may
not be possible in other studies. The following details an example for two data streams.
Step 1: Assimilation of the first data stream, $s_1$. The prior parameters, including their values

22          and error covariance ($\mathbf{x}^b$ and $\mathbf{B}$), are optimised to produce a first set of posterior

23          optimised parameters $\mathbf{x}_1$ with error covariance $\mathbf{A}_1$.

Step 2:  Assimilation the second data stream, $s_2$. The parameters, $\mathbf{x}_1$, and their error

25          covariance, $\mathbf{A}_1$, are used as a prior to the optimisation system and further optimised

26          to produce the second (and final) set of posterior optimised parameters, $\mathbf{x}_{post}$, and the

27          associated error covariance $\mathbf{A}$.





*Simultaneous approach*
Both data streams $s_1$ and $s_2$ are included in the optimisation and all parameters are optimised
at the same time. The prior parameters, including their values and error covariance ($\mathbf{x}^b$ and $\mathbf{B}$)
are optimised to produce the posterior parameter vector ($\mathbf{x}_{post}$) and associated uncertainties $\mathbf{A}$.
### 2.1.5 Optimisation set-up: parameter values and uncertainty, and generation

7           of synthetic observations

8          In this study we used synthetic observations that were generated by running the model

with known (or 'true') parameter values and adding random Gaussian noise corresponding to
the defined observation error for both $s_1$ and $s_2$ (see Table 1). The true values of all parameters
for both models are given in Table 1, together with their upper and lower bounds. The
parameter uncertainty (1 sigma) was set to 40% of the parameter range following recent
studies (e.g. Bacour et al., 2015). Prior values were chosen from a uniform random
distribution bounded by the parameter bounds. We assumed independence (i.e. uncorrelated
errors) for both the parameters and observation covariance matrices, thus the $\mathbf{R}$ and $\mathbf{B}$
matrices were diagonal.
### 2.1.6 Experiments

19         Table 2 details the experiments that were carried out based on all possible combinations

for assimilating the two data streams. Three approaches were compared: i) separate – where
only one data stream was included in the optimisation; ii) step-wise – where each data stream
was assimilated sequentially; and ii) simultaneous – where both data streams were included in
the optimisation. All parameters for both models were optimised in all experiments, therefore
in the step-wise cases the parameters were optimised twice. Tests for the step-wise were also
carried out with and without the propagation of the full posterior parameter error covariance
matrix, $\mathbf{A}_1$, in between steps 1 and 2 (test cases 2b and d – see Table 2) – i.e. for these tests
only the posterior variance was propagated. An additional test was included for the
simultaneous assimilation in order to test the impact of having a substantial difference in the
number of observations for the data stream included in the optimisation; therefore in test case
3b, only one observation was included for data stream $s_2$.





The differences in the parameter values and the theoretical reduction in their uncertainty
($1 - (\sigma_{post} / \sigma_{prior})$) were examined for all eight test cases, as well as the fit (RMSE) to both
data streams after the optimisation. For the step-wise approach we investigated if the fit to the
first data stream is degraded in the second step by comparing the RMSE after each step. Note
that the reduction in uncertainty is a theoretical, or approximate, estimate of the real
uncertainty reduction because of the assumptions made in the inversion scheme.
In a second stage the impact of an unknown, un-accounted for bias in the model was
examined. This bias could be a systematic bias in the observations due to the algorithm used
for their derivation, the result of missing or incomplete processes in the model, or an
incompatibility between the observations and the model, for example due to differences in
spatial resolution or an inconsistent characterisation of a variable between the model and the
observations. To test the impact of such an occurrence, we introduced a uniform scalar bias
into the modelled $s_2$ variable with a value of 10 (i.e. twice the magnitude of the defined
observation uncertainty). All eight experiments were repeated, but a bias was introduced into
the model calculation of $s_2$ that was not accounted for in the cost function (i.e. the error
distributions retained a mean of zero). This was treated as an unknown bias, and therefore not
corrected or accounted for in the inversion scheme and the defined observation uncertainty
(Table 1) was not changed for this set of experiments.
In all experiments for both models twenty assimilations were performed starting from
different random "first guess" points in the parameter space. As discussed in Section 2.1.3
this was done to test the ability of the algorithm to converge to the true global minimum of the
cost function. Note that the global minimum and possible reduction in $J(x)$ will be different
for each experiment, as each is based on a different cost function.

## 2.2   Results

The twenty random first guess assimilations were examined for each set of experiments
for both models (before the results for each test were examined in more detail), in order to
check that the algorithm converged to a global minimum. As shown in the supplemental
information (Fig. S1), a high proportion of the twenty first guess assimilations across all test
cases for both models resulted in a similar reduction in $J(x)$, even though the overall
magnitude of the reduction was sometimes different between tests. This indicates that the





algorithm does not easily get stuck in any local minima (if they exist). The examples shown in
the results below were taken from one first guess parameter set for each model that belonged
to the cluster that had the highest cost function reduction. Any differences seen in the
parameter values, their posterior uncertainty or the resultant RMSE reduction described below
therefore are due to the specific details of each test and not the inability of the algorithm to
find the minimum.
### 2.2.1  Typical performance with a quasi-linear model and no bias
Figures 1a and b show the simple carbon model simulations for test case 3a (in which
both data streams are assimilated simultaneously) for the $s_1$ and $s_2$ variables. A large reduction
in RMSE is achieved after optimisation (blue curve) with respect to the observations (black
curve). Overall, there is a good reduction in RMSE for all test cases (including the individual
assimilations 1a and 1b) with a reduction of ~80% for $s_1$ and $s_2$. In addition, the optimisation
of the $s_1$ and $s_2$ variables resulted in a good or moderate reduction in RMSE for variables not
included in any assimilation: ~60% for the litterfall (Eqn. 1) and ~16% for the heterotrophic
respiration (Rh – Eqn. 2) across all test cases (not shown), although there was already a good
prior fit to the data. As would be expected from these results, the parameter values and the
theoretical reduction in parameter uncertainty do not vary between the tests (Figures 2a and b
blue symbols), except for a slight difference in the value of the $k_2$ parameter in test cases 1a
and 3b, for which there is also a lower reduction in uncertainty (~82% compared to >95%).
Note that Fig. 2a shows the normalised parameter values to account for differences in the
magnitude of the different parameters and their range (the zero line represents the "true"
parameter value – see caption). In this situation therefore, where we have a relatively simple
linear model and two data streams to which the model parameters are highly sensitive, we see
that the differences between the step-wise and simultaneous approaches are minimal. This is
even the case when the error covariance is not propagated between the two steps (test cases 2b
and d), suggesting that under this assimilation set-up both $s_1$ and $s_2$ individually contain
enough information to retrieve the true values of all parameters, as we can see from the
separate test cases 1a and b.



### 2.2.2 Impact of unknown bias in one data stream – example with a simple carbon model

In Section 2.2.1 we saw that there is little difference between a step-wise and simultaneous optimisation if there is no bias in the model or observations, and if the model is quasi-linear and therefore the critical assumptions behind the inversion approach were not violated. However, it is not uncommon to have a bias between your observations and model that is not obvious and therefore not accounted for in the optimisation, as the cost function used in most inversion algorithms (and in this study) assume Gaussian error distributions with zero mean. Note that this is also the case when defining a likelihood function for accepting or rejecting parameter values in a global search method. To test the impact of a bias, we added a uniform value to the simulated $s_2$ variable in a second test (see Section 2.1.6) that was treated as an unknown bias, and therefore not corrected or accounted for in the inversion scheme. The impact of this bias on $s_1$ and $s_2$ is shown in Figures 1c-d, and the reduction in RMSE between the model and observations is seen in Fig. 3 for all variables (including Rh and litterfall). The red symbols in Fig. 2 show the resultant parameter values and theoretical reduction in uncertainty as a result of the bias. The inversion cannot accurately find the correct values for all parameters in any test case and there are now considerable differences between the simultaneous and step-wise approach. Furthermore the order in which the data streams are assimilated in the step-wise cases also results in different posterior parameter values (test cases 2a and b versus 2c and d in Fig. 2a and Fig. 3). Nevertheless the optimisation results in a similar reduction in uncertainty on the parameters, except in test case 1b where only $s_2$ data are assimilated (Fig. 2b).

The main impact of the bias in the modelled $s_2$ variable is on the value of $k_2$ parameter (Fig. 2a), which is consistently offset from the true value (dashed line in Fig. 2a) in all test cases. This was expected given that it is the parameter most directly related to the calculation of $s_2$. However, in test cases 2a and 3a, the values of $p_1$ and $p_2$ are also incorrect (and $p_1$ for test case 2b). Note that these parameters only indirectly influence the $s_2$ pool in the model, and therefore we might have expected that they would be less affected by the bias. This nicely demonstrates one issue that could arise in all DA studies, where the bias in a particular variable (in the observations or the model) is aliased onto another process in the model (Wutzler and Carvalhais, 2014). Such an aliasing of bias onto indirectly related parameters is even more evident when only $s_2$ is included in the assimilation and $s_1$ does not provide any





constraint (test case 1b) – in this case all parameters are incorrect but the $p_2$ parameter in
particular shows a strong deviation from the true value (Fig. 2a). As a result we see a
deterioration in the RMSE for the $s_1$, litterfall and Rh variables in test case 1b and in the step-
wise cases where $s_2$ is assimilated in the second step (Figures 3a, c and d – test case 1b, 2a
and 2b). However, the RMSE reduction remains high for the $s_2$ variable for these test cases
(Fig. 3b), as the inversion has found a solution that accounts for the bias even though all
inferred parameter values are incorrect. The assimilation of $s_1$ in the second step lowers the
reduction in RMSE for $s_2$ gained in the first step to ~70%, but it is not a considerable
degradation.
Even though the posterior parameter values are incorrect, and despite the fact that the
first step results in a degradation, the final reductions in RMSE are largely the same than the
situation with no bias for all variables when $s_1$ is included in a simultaneous assimilation or
optimised in the second step (test cases 2c, d and 3a in Fig. 3). This shows that the inclusion
of $s_1$ observations can find a solution to counter the bias in $s_2$ and prevents a degradation in
the fit to the data. If $s_2$ is assimilated in the second step there is a negative impact on all other
variables as discussed above, demonstrating again that the order of data stream assimilation
can matter when there are biases or inconsistencies between the data and the model.
The analysis of the impact of the bias presented here is specific to this model and the
type and magnitude of the bias that was added, but the broader findings can be generalised to
any situation in which there is a bias or inconsistency between a model and data that is not
accounted for in the assigned error distributions. Exactly what might constitute a bias or
inconsistency is discussed more in Section 3.2. Also note that it is important to examine the
impact on the other variables. For the separate test case 1b in which only $s_2$ data are used to
optimise the model, the negative impact on the other variables (Fig. 3) would have been
concealed if we had only examined the posterior reduction in RMSE for the $s_2$ variable. Again
this is a concern that is inherent to all DA experiments, whether single- or multi-data stream,
but we can see from these results (i.e. by comparing the separate test cases 1b with 2a and b)
that adding another data stream in a multi-constraint approach does not always reduce the
problem.





### 2.2.3 Difference between the step-wise and simultaneous approaches in the presence of a non-linear model

As discussed in Section 2.2.1, there is little difference between the step-wise and the simultaneous assimilation approaches for simple, relatively linear models, unless the observation error (including measurement and model errors) distribution deviates strongly from the Gaussian assumption. However in reality, large-scale, complex LSMs may contain highly non-linear responses to certain model parameters. To demonstrate the impact of non-linearity in a multiple data stream assimilation context, we used a non-physically based toy model chosen for its non-linear characteristics (see Section 2.1.2).

Fig. 4a shows the posterior parameter values for both the $a$ and $b$ parameters of the non-linear toy model for all test cases. The values were not normalised as both parameters have the same range. The horizontal dashed line shows the "true" known values of the parameters (both equal to 1.0) that were used to generate the synthetic observations. Note that no bias has been introduced into the model in the results described here. The prior and posterior model $s_1$ and $s_2$ simulations for the non-linear toy model are compared to the synthetic observations in Fig. 5 for both step-wise cases in which the posterior error covariance matrix from step 1 ($A_1$ – see section 2.1.4) was propagated to step 2 (experiments 2a and c – Fig. 5a-d) and both simultaneous cases 3a and b (Fig. 5 e-h). Finally Fig. 6 summarises the reduction in RMSE between the simulated and observed $s_1$ and $s_2$ variables for the non-linear toy model for all test cases and, in the step-wise cases, the reduction in RMSE after both the first and second steps (light versus dark green bars).

Assimilating each data stream individually (test cases 1a and b) does not result in an accurate retrieval of the posterior parameters (Fig. 4a), nor in a strong constraint on either parameter, as shown by the lack of theoretical reduction in the parameter uncertainty after the optimisation (Fig. 4b). Despite this, there is a 91-92% reduction in RMSE for the data stream that was included in the optimisation (i.e. for $s_1$ in test case 1a – Fig. 6a, and $s_2$ in test case 1b – Fig. 6b). However, the improvement on the other data stream is much less (28% reduction in RMSE for $s_1$ when $s_2$ is assimilated) or even results in a degradation compared to the prior fit (e.g. in the case of $s_2$ when $s_1$ is assimilated – Fig. 6b). Lack of improvement, or even degradation, in the RMSE of other variables in the model is a common issue for data assimilation in general – one that is not often evaluated in model-data fusion studies.



Only the simultaneous case, in which all $s_1$ observations have been included in the cost
function (test case 3a), manages to retrieve the correct parameter values after the optimisation.
All other posterior parameter values are incorrect, and are considerably different between
each case, unlike for the simple carbon model (without a model bias). Most step-wise test
cases (particularly 2b-d) do not result in the same parameter values as the simultaneous test
case 3a in which all the observations are included (Fig. 4a), highlighting that strong non-
linearity in the model sensitivity to parameters together with the use of an algorithm that is
only adapted to weakly non-linear problems, as well as the assumption of linearity in
calculating the posterior error covariance matrix at the minimum of the cost function, can
result in differences between a step-wise and simultaneous approach in multiple – data stream
assimilation (see Section 1).
In the simultaneous optimisation in which all observations are included (test case 3a)
the posterior fit to the data dramatically improves for both the $s_1$ and $s_2$ data streams after the
assimilation (blue dashed line in Fig. 5e and f). This was expected given that the correct
values of the parameters were found. For the step-wise cases (test case 2a in Figures 5a and b,
and test case 2c in Fig. 5c and d), the black dashed line shows the prior, and the posterior after
step 1 is shown by green dashed line. In the step-wise assimilation we see two different
scenarios depending on which data stream was assimilated first. In the first step the results are
the same as the case where each individual data stream is assimilated separately. In both cases
the first step results in a good fit to the data that was included in the optimisation in that step.
When the $s_1$ data was assimilated in the first step (Fig. 5 first row), the fit to $s_2$ deteriorated
after the optimisation (Fig. 5b green dashed line and Fig. 6b – test case 2a_s1), but when the
$s_2$ data were assimilated first (Fig. 5 second row) the optimisation step did manage to achieve
an improvement in the $s_1$ data stream (Fig. 5c green dashed line and Fig. 6a – test case 2c_s1).
In the second step the optimisation of $s_2$ in test cases 2a and b does not degrade the fit
to $s_1$ when the full parameter error covariance matrix ($\mathbf{A}_1$) is propagated between step 1 and 2
(Figures 5a blue curve and 6a 2a_s2). Furthermore optimising $s_2$ in the second step reverses
the deterioration in $s_2$ caused by assimilating $s_1$ in the first step (Figures 5b blue curve and 6b
2a and b dark green bars). However, when $s_1$ data were assimilated in the second step (test
cases 2c and d), we found that the good fit achieved with $s_2$ observations in the first step was
effectively reversed (Fig. 5d blue curve). Therefore assimilating $s_1$ in the second step
degraded the fit to the $s_2$ observations, even compared to the prior case (Fig. 6b, dark green



bars for test cases 2c and d). This nicely highlights one of the main possible issues with a
step-wise assimilation framework.
The fact that the final reduction in RMSE values after both steps was ~90% for most
cases, even though the values were not correct for all but case 3a (Fig. 4), indicates that the
error correlation between the two parameters (~ -1.0 – calculated from the posterior error
covariance matrix but not shown) led to alternative sets of values that resulted in a similar
improvement to the data – a phenomenon known as model equifinality.

### 2.2.4 Order of assimilation of data streams and propagation of parameter error covariance matrices in a step-wise approach

Comparing the step-wise cases 2a and b with 2c and d for the non-linear toy model
reveals that neither order in the assimilation, $s_1$ then $s_2$, or $s_2$ then $s_1$, results in the correct
posterior parameter values that match the simultaneous test case (Fig. 4a). This is not a result
that can be generalised to all step-wise assimilations as it will depend on the data stream
involved and whether they contain enough information to accurately constrain all the
parameters included in the optimisation, as well as any biases in the model or observations (as
discussed in Section 2.2.2) or model non-linearity (section 2.2.3). In the case of the non-linear
toy model, neither $s_1$ nor $s_2$ find the right parameter values when assimilated individually,
therefore it is not surprising that neither order manages to achieve the right posterior
parameter values. Nevertheless, the theoretical uncertainty of both parameters is reduced by
>95% for the step-wise cases in which $\mathbf{A}_1$ from step 1 is propagated between step 1 and 2 (test
cases 2a and c – Fig. 4b), even though the posterior values for the step-wise cases are
incorrect. This demonstrates that a good theoretical reduction in uncertainty is not always
indicative that the right parameters have been found by the optimisation. The lower
theoretical reduction in parametric uncertainty for cases 2b and d (Fig. 4b) demonstrates that
information is lost between the steps if the posterior error covariance terms of $\mathbf{A}_1$ after step 1
are not propagated to step 2, and therefore cannot be used to further constrain the
optimisation.
From a mathematical standpoint the most rigorous approach is to propagate the full
parameter error covariance matrices between each step. Without that constraint not only is
information lost in the second step, but the information contained in the second data stream





may have a stronger influence compared to a simultaneous or step-wise case with a
propagated error covariance matrix. The inversion may therefore be more vulnerable to any
strong biases or incompatibilities between the model and the observations of the second data
stream, or indeed the particular sensitivity of its corresponding model state variable to the
parameters. This is one possible explanation for the degradation seen in $s_1$ in the non-linear
toy model when $s_2$ is optimised in the second step and $\mathbf{A}_1$ is not propagated between the steps
(Fig. 6a test case 2b_s2). The same was also true for the simple carbon model for test case 2b
when a bias was introduced into the $s_2$ simulation (see Section 2.2.2 and Fig. 3a).
However, the reverse is also true – if the first data stream contains strong biases then
the associated error correlations will be also propagated with $\mathbf{A}_1$. If autocorrelation in the
observation errors, or indeed correlation between the errors of the data streams, is not
accounted for it is likely that the posterior simulations are over-tuned, i.e. we will
overestimate the reduction in parameter uncertainty. If this is the case and the first step results
in incorrect parameter values, the propagation of $\mathbf{A}_1$ could restrict the parameter values to the
wrong location in the parameter space and thus inhibit the ability of the inversion to find the
correct global minimum. These issues are likely to be more considerable for non-linear
models, as seen by the lack of difference between test cases 2a-d in the simple carbon model
example (Fig. 2).

## 20    2.2.5  Lessons to be learned when dealing with non-linearity

Most optimisation studies with a large-scale LSM use derivative methods based on a
least-squares approach, and therefore rely on assumptions of Gaussian probability and linear
model sensitivity. However, if the model is weakly non-linear within the probability
distribution around the point in parameter space that is being analysed (see Tarantola, 1987,
p72), it is possible to use an iterative algorithm, such as the one described in Eq. (6), to find
the minimum of the cost function (i.e. the maximum likelihood of the posterior parameter
distribution). Furthermore a linearization of the model around the maximum likelihood
estimation (minimum of $J(\mathbf{x})$) of the parameters can be used to calculate the posterior error
covariance (see Eq. (6)). If the model is too strongly non-linear and therefore these
assumptions are not met, it may not be possible to find the true global minimum of the cost
function and the characterisation of the posterior probability distribution will be incorrect.





This is a particular problem if the posterior parameter error covariance matrix is then
propagated in a step-wise approach, although these issues are relevant to both step-wise and
simultaneous assimilation. Note that performing a number of tests starting from different
random "first guess" points in parameter space can help to diagnose if the global minimum
has been reached, as outlined in Section 2.1.6 and discussed at the beginning of the results
(Section 2.2).
It is possible to avoid dealing with issues related to non-linearity in the model
sensitivity and non-Gaussian error distributions by using a global search method (e.g. Markov
Chain Monte Carlo or a genetic algorithm) that randomly, but effectively, searches the entire
parameter space. However in large dimensional problems, as is likely the case when
optimising a LSM at large scales with multiple data streams, using a global search method is
likely not feasible due to computational time constraints. In these cases, a derivative method
is likely the only option.
An important finding of the results presented for the non-linear toy model in Section
2.2.3 is that degradation in another data stream is not necessarily the result of a bias or
incompatibility between the observations and the model. Rather if the model sensitivity to the
parameters is very non-linear, multiple combinations of parameter values may exist that result
in a similar reduction of the cost function (multiple minima), but provide a different fit to
each data stream. Even though all data streams may be sensitive to all the parameters, the
information content of each will not be the same. Finding the true global minimum in this
instance may require a bit more careful thought in planning the assimilation set-up, and may
depend on having a reasonable idea of the level of information each data stream can bring to
constrain each parameter. It may be the case that one data stream has a higher non-linear
sensitivity to the parameters and therefore may act as an "troublemaker" and pull the
parameters in a direction that results in a degradation to the other data streams, as seen in
Section 2.2.3. If a simultaneous optimisation is not possible, it may be useful under such
circumstances to identify any "troublemaker" data streams, and assimilate them in the first
step of the optimisation. In the second step "peacemaker" data streams, with a lower non-
linear sensitivity to the parameters, will then find a compromise set of parameter values that
can fit both data streams well, provided the full posterior parameter error covariance matrix is
propagated between the steps in order to retain all the information brought by the first data
stream. As discussed this could be an explanation for the results seen for the non-linear toy



model test case 2a where $s_1$ was assimilated prior to $s_2$ (Figures 6a and b) as discussed in
Section 2.2.3.

## 3 Examples from existing carbon cycle data assimilation studies

### 3.1 Extra constraint from multiple data streams

Most site-based carbon cycle data assimilation studies have used eddy covariance
measurements of NEE and LE fluxes to constrain the relevant parameters of ecosystem
models. However, a few studies have also made use of chamber flux soil respiration data and
field measurements of vegetation characteristics (e.g. tree height, budburst, LAI) or estimates
of litterfall and carbon stocks as ancillary information (e.g. Keenan et al., 2012; Thum et al.,
in review; Van Oijen et al., 2005; Richardson et al., 2010; Williams et al., 2005). Two recent
studies combined high-resolution satellite-derived FAPAR data and in-situ eddy covariance
measurements to optimize parameters related to carbon, water and energy cycles of the
ORCHIDEE and BETHY LSMs (Bacour et al., 2015; Kato et al., 2013, respectively).
At global scales the number of studies that use multiple data streams from satellites or
large-scale networks to optimise LSMs has been increasing in recent years, although this
remains a relatively new area of research. CCDAS-BETHY was the first global carbon cycle
data assimilation system (CCDAS) making use of the high-precision measurements of the
atmospheric $CO_2$ concentration flask sampling network (Rayner et al., 2005; Scholze, 2003)
to constrain process parameters of the prognostic terrestrial carbon cycle model BETHY
(Knorr, 2000). Since its first application assimilating atmospheric $CO_2$ concentration data
only, CCDAS-BETHY has been further developed to consistently assimilate multiple data
streams both at local and global scales. In particular, Kaminski et al. (2012) optimised 70
process parameters plus one initial condition by simultaneously assimilating a satellite-
derived FAPAR product derived from the Medium Resolution Imaging Spectrometer
(MERIS; Gobron et al., 2008) and flask samples of atmospheric $CO_2$ at two sites from the
GLOBALVIEW product (GLOBALVIEW-CO2, 2008) on a coarse resolution. More recently,
Scholze et al. (2015) demonstrated the added value of assimilating remotely sensed soil
moisture data in addition to observations of atmospheric $CO_2$ concentration from the flask-
sampling network. They used the same coarse resolution set-up of CCDAS as Kaminski et al.
(2012) and $CO_2$ observations from 10 sites of the GLOBALVIEW product (GLOBALVIEW-



CO2, 2012) together with the SMOS L3 daily soil moisture product (version 246; CATDS-
L3, 2012).

3         Two other global CCDAS based on different LSMs have been developed in recent years

(Peylin et al., 2016; Schürmann et al., 2016). Schürmann et al. (2016) optimized model
parameters and initial conditions of the land component JSBACH (Raddatz et al. 2007) of the
MPI Earth System Model (ESM) (Giorgetta et al. 2013) using atmospheric $CO_2$ concentration
data and the TIP-FAPAR product (Pinty et al., 2007) as joint constraints over a 5 year period
in addition to evaluating the mutual benefit of each data stream in a fully factorial design.
Peylin et al. (2016) used three different data streams as global constraints for the ORCHIDEE
LSM (Krinner et al., 2005), which forms the land surface component of the IPSL ESM
(Dufresne et al., 2013), in a multi-site step-wise assimilation approach. First, satellite-derived
vegetation index data (NDVI) from the MODIS instrument was used to constrain the
phenology parameters at 60 sites for the temperate and boreal deciduous PFTs, followed by
NEE and LE observations at 78 FLUXNET sites for 7 PFTs to optimise all the carbon-related
parameters, and finally atmospheric $CO_2$ concentration measurements from 53 sites in the
GLOBALVIEW network (GLOBALVIEW-CO2, 2013), which predominantly provided a
constraint on the initial magnitude of the soil carbon reserves in the model. Atmospheric $CO_2$
concentration observations are one of the most accurate, long-term data sets in environmental
science and they provide important information about the global $CO_2$ sink capacity by land
and ocean. These three global multiple data stream CCDAS have allowed an improvement in
both the mean seasonal cycle as well as the trend of net land surface $CO_2$ exchange, especially
with the inclusion of the atmospheric $CO_2$ data (Kaminski et al., 2012; Peylin et al., 2016;
Schürmann et al., 2016).

24        Many of the aforementioned studies reported that adding extra data streams helped to

constrain unresolved sub-spaces of the total parameter space. Scholze et al. (2015) found that
adding SMOS soil moisture data to the assimilation simultaneously with $CO_2$ observations
reduced the ambiguity in the solution space when assimilating $CO_2$ only, and the multiple
data constraint was able to resolve a much larger sub-space in parameter space (about 30
parameters out of the 101 compared to 15 without SMOS data). Bacour et al. (2015) and
Schürmann et al. (2016) both reported that the addition of FAPAR data bought extra
information on the phenology-related processes in the model, and therefore retrieved different
posterior C flux-related parameter values than when assimilating NEE or atmospheric $CO_2$



data alone. An interesting aspect of the Kaminski et al. (2012) study was that the inclusion of
FAPAR in addition to atmospheric $CO_2$ concentration samples resulted in a particular
improvement for the hydrological fluxes in the model, thus demonstrating the importance of
assessing the potential benefit for model variables that may not have been the main target of
optimisation. Richardson et al. (2010) concluded that using ancillary information (e.g. woody
biomass increment, field-based LAI and chamber measurements of soil respiration) as
orthogonal constraints to NEE data provided a valuable added constraint on many model
parameters, which improved both the bias in model predictions and reduced the associated
uncertainties. Thum et al. (in review) also found that the addition of aboveground biomass
stocks brought a longer-term constraint on allocation parameters and mortality/turnover
processes. However, they noted an incompatibility when assimilating both annual increment
and total biomass data, as the total stocks take into account losses related to disturbance and
management (e.g. canopy thinning) – processes that were not included in that version of the
model. Keenan et al. (2012) also argued that the use of such ancillary constraints is essential
for a better partitioning of net carbon fluxes into their gross components. However, Williams
et al. (2005) observed that one-off, or rarely taken, measurements of carbon stocks were
unable to constrain components of the carbon cycle to which they were not directly related.
This calls into question the issue of the influence of different data streams in a joint
assimilation, especially if the number of observations for each is vastly different which is the
case when assimilating both half-hourly C flux data in addition to soil C stock observations
that are typically available at an annual time scale. The spatial coverage of each data stream is
also important, especially for heterogeneous landscapes (Barrett et al., 2005). Test case 3b, in
which only one observation was included for the $s_2$ data stream instead of the complete time-
series, shows that a substantial difference in number of observations between the data streams
can influence the resulting parameter values and posterior uncertainty (compare test cases 3a
and b in Fig. 2 for the simple C model and Fig. 4 for the non-linear toy model) as each data
stream will have a different overall "weight" in the cost function. However, the impact of
having a different number of observations for each data stream in the cost function also
depends strongly on the prescribed observation error and relative sensitivity of each
corresponding model variable to the model parameters. If one variable has a greater
sensitivity than the other, it will matter less if fewer observations of that variable are included
in the cost function.





Xu et al. (2006), among others, have mentioned the possible need to weight the cost
function for different data sets. Different arguments abound on this issue. Some contend that
the cost function should not be weighted by the number of observations because the error
covariance matrices (**B** and **R**) already define this weight in an objective way (e.g. Keenan et
al., 2013). Certainly it should not be necessary to weight by the number of observations in the
cost function if there is sufficient information to properly build the prior error covariance
matrices (**B** and **R**). On the other hand, it is a difficult task to characterise the model structural
uncertainty and the observation error correlations (see Kuppel et al., 2013 for practical
solutions). Given this, our expert knowledge on the workings of the model processes and the
sensitivity of the model to the parameters may permit us to specify a stronger weight to a data
stream that could help to constrain a particular section of the model, but for which there are
only a few data points. Clearly the definition of the prior error model, including for the
covariance between errors of the data streams, is of the upmost importance (Trudinger et al.,
2007) and merits close attention in future multiple data stream assimilation studies.
Although a number of multiple data stream assimilation studies exist at various scales,
very few studies have specifically investigated the added benefit of different combinations of
data streams, with a few notable exceptions (Barrett et al., 2005; Richardson et al., 2010; Kato
et al., 2013; Keenan et al., 2013; Bacour et al., 2015; Schürmann et al., 2016). Kato et al.
(2013) and Bacour et al. (2015) both evaluated the complementarity of eddy covariance and
FAPAR data streams at site level, i.e. the impact of assimilating one individual data stream on
the other model state variable, as well as when both data streams were included in the
optimization (see discussion in Section 3.2). The study of Keenan et al. (2013) was
particularly notable in its aim to quantify which data streams provide the most information
and how many data streams are actually needed to constrain the problem. They reported that
of the 17 field-based data streams available, projections of future carbon dynamics were well-
constrained with only 5 of the data sources, and crucially, not with eddy covariance NEE
measurements alone. These results may be specific to this site or type of ecosystem, but this
study highlights the need for further research in this area, and in particular, for synthetic data
experiments that allow us to understand which data will be the most useful for a given
scientific question. This will also enable researchers to plan more efficient measurement
campaigns with experimentalists, as also pointed out by Keenan et al. (2012).



## 3.2  Issue of bias and inconsistencies between the observations and the model

Despite the theoretical benefit of adding data streams into an assimilation system as orthogonal constraints, several of the aforementioned studies at both site and global scale have reported a bias or inconsistency either between the different observation data streams, or between the observations and the model. This is easily detected when the optimisation of one data stream results in a worse fit than the prior in one or more of the other data streams, as seen in Section 2.2.2. Kato et al. (2013) assimilated SeaWiFS FAPAR (Gobron et al., 2006) and eddy covariance LE measurements at the FLUXNET site in Maun, Botswana. They showed that the individual assimilation of each the two data streams resulted in a perfect (i.e. within the observational uncertainty) fit to the assimilated data set, but a considerable degradation of the fit to the non-assimilated data set compared to the prior. A comparison against eddy covariance measurements of gross carbon uptake (gross primary production – GPP) hinted to a bias problem with the FAPAR data because the fit to the independent GPP data was degraded after assimilating FAPAR data only, while the fit improved after assimilating the LE data only. Nevertheless, the simultaneous assimilation of both data streams achieved a compromise between the two suboptimal states reached after assimilating only one data stream. The calibration further limited the number of parameters with correlated errors, and yielded a higher theoretical reduction in parameter uncertainty and a decrease in the RMS difference by 16% for the GPP data compared to the prior.

Bacour et al. (2015) also noted that when assimilating both in-situ and satellite-derived FAPAR data (from the SPOT and MERIS instruments) and in-situ NEE and LE flux data from two French FLUXNET sites into the ORCHIDEE LSM both separately and together, the posterior parameter values changed significantly for the photosynthesis and phenology-related parameters, depending on the bias between the model and the observations and the correlation between the parameter errors. If NEE data were assimilated alone there was an even stronger positive bias (model–observations) in the start of leaf onset in the FAPAR data than in the prior simulations, and no improvement in the maximum value. This was likely due to the fact that there were enough degrees of freedom to fit the NEE without changing the phenology-related parameters. Similarly, the fit to the NEE was degraded when the model was only optimized with FAPAR data. The model was able to fit the maximum FAPAR but this resulted in an adverse effect on the carbon assimilation capacity of the vegetation. The





authors argued this was related to incompatibilities between the FAPAR and both the model
and NEE measurements, possibly due to its larger spatial footprint of the satellite-derived
FAPAR data and/or inaccuracies in the retrieval algorithm. However, given that assimilating
in-situ FAPAR also degraded the fit to the NEE, another culprit may be an inconsistency
between the model and the data. The authors suggested this could be due to the different
assumptions or characterisation of a variable in a model compared to what is described in the
data. For example, satellite-derived greenness measures (FAPAR/NDVI) also contain
information on the non-green elements of vegetation, but the model only simulates green LAI.
Furthermore parameters and processes in models have been developed at certain temporal and
spatial scales. Vegetation is often simply represented as a "big leaf" model in LSMs, taking
no account of vertical canopy structure or the spatial heterogeneity in a scene, which is an
additional source of inconsistency with what is measured. The joint (simultaneous)
assimilation of all three data streams in Bacour et al. (2015) reconciled the different sources
of information, with an improvement in the model-data fit for NEE, LE and FAPAR.
However, the compromise achieved in the joint assimilation was only possible when the
FAPAR data were normalised to their maximum and minimum values, which thus partially
accounted for any bias in the magnitude of the FAPAR or inconsistency with the model.
The story of biases and apparent inconsistencies in FAPAR data doesn't end there. A
bias correction was also necessary in the study by Kaminski et al. (2012) with CCDAS-
BETHY using the MERIS FAPAR product in addition to atmospheric $CO_2$ data (see above).
They found that optimisation procedure failed when using the original FAPAR product
because the FAPAR values were biased towards higher values. Only after applying a bias
correction on the FAPAR data before the assimilation procedure was the optimisation
successful. Schürmann et al. (2016) also reported the need to reduce a prior model bias in
FAPAR. Even though the assimilation corrected successfully for this FAPAR bias, an imprint
of the prior bias was evident in the spatial patterns of the modelled heterotrophic respiration.
Assimilating FAPAR data alone therefore resulted in a slight degradation in the net C flux
and consequently led to incorrect simulations of the atmospheric $CO_2$ growth rate. The
addition of $CO_2$ as a constraint prevented this degradation and resulted in a compromise in
which FAPAR helped to disentangle these processes and find different parameter values
compared to the $CO_2$-only case, thus improving the fit to both data streams. Forkel et al.
(2014) discovered an apparent inconsistency between satellite-derived FAPAR and GPP data
in tundra regions when using these data (plus satellite-derived albedo) to optimise the LPJmL





1 LSM. They too speculated that the data might be positively biased, in this case due to issues

2 with satellite measurements taken at high sun zenith angles. However, they gave alternative

3 suggestions, one being that an inadequate model structure may be at fault – for example, the

4 LPJmL does not include vegetation classes corresponding to shrub, moss and lichen species

5 that are dominant in these ecosystems. They also noted that the GPP product they used, which

6 is based on a model tree ensemble up-scaling of FLUXNET data (Jung et al., 2011), might

7 contain representation-related biases, given that there are very few FLUXNET stations in

8 tundra regions. The issue of representation errors of sites has been touched upon before (e.g.

9 Raupach et al., 2005). Alton (2013), who performed a global multi-site optimisation of the

10 JULES LSM with a diverse range of data including satellite-derived LAI, FLUXNET, soil

11 respiration and global river discharge, raised the point that FLUXNET sites are known to be

12 large carbon sinks, which could potentially result in biased global NEE estimates. Resolving

13 these apparent inconsistencies was beyond the scope of most of these studies, aside from

14 applying a bias correction where one was evident. Nonetheless this issue clearly merits further

15 attention if the increasing number of available datasets is to be fully utilised.

## 3.3 Step-wise versus simultaneous assimilation

18  The paper by Alton (2013) documents the only previous study to have used a step-wise

19 assimilation approach with more than two data streams, stating that the final parameter values

20 were independent of the order of data streams assimilated. No studies in the LSM community

21 to date have explicitly examined a step-wise versus simultaneous assimilation framework

22 with the same optimisation system and model. The step-wise assimilation with the

23 ORCHIDEE-CCDAS detailed in Peylin et al. (2016) has been compared to a simultaneous

24 optimisation using the same three data streams as part of an on-going study. At each step, the

25 resulting simulations (using the posterior parameters) were compared to the data stream from

26 the previous steps. The fit to the MODIS NDVI (used in a similar manner to FAPAR as a

27 proxy for vegetation greenness) was unchanged after further optimization of the phenology-

28 related parameters in the second and third steps using in-situ flux and atmospheric $CO_2$

29 concentration data. In the simultaneous optimisation, the addition of NEE or atmospheric $CO_2$

30 concentration measurements resulted in a lower improvement to the fit to MODIS NDVI. As

31 the NDVI data were normalised this was not a result of a simple bias in the magnitude of the

32 data. Rather, it was likely due to inconsistencies between the model and data as discussed by




Bacour et al. (2015) and in Section 3.2. It is important to reiterate that there should be no difference between the step-wise and the simultaneous given an adequate description of the error covariance matrices and compliance with the assumptions associated with the inversion algorithm used. However, in practice it is very difficult to define a PDF that properly characterises the model structural uncertainty and observation errors accounting for biases and non-Gaussian distributions. This leads to issues within a simultaneous assimilation, particularly if the information content of one data stream is much higher, and a greater risk of differences between a step-wise and simultaneous assimilation. As discussed in Section 2.2.5 a step-wise assimilation may be useful on a provisional basis for dealing with possible inconsistencies. In the step-wise approach of Peylin et al. (2016) the error covariance of the phenology-related parameters was strongly constrained by the satellite data in the first step (and was propagated to the second step), the later assimilations with NEE and atmospheric $CO_2$ data in steps 2 and 3 found alternative solutions for the C flux-related parameters that provided a reasonable fit to all data streams. Wherever possible however, a simultaneous optimisation is favourable because the strong parameter linkages between different processes are maintained, and therefore biases and inconsistencies between the model and observations should be addressed prior to optimisation.

## 4    Advice for Land Surface Modellers

Although it is clear that in many cases, increasing the number of observations in a model optimisation provides additional, orthogonal constraints, challenges remain that should not be ignored. Based on the simple toy model results presented in this study, in addition to lessons learned from existing studies, we recommend the following points when carrying out multiple data stream carbon cycle data assimilation experiments:

- Devote time to characterising the error structure for the observations and parameter error distributions, including their correlations (Raupach et al., 2005). For the observations this should include the model structural errors (Kuppel et al. 2013), the temporal or spatial autocorrelation and correlation between different data streams.

- In the case of non-Gaussian error distributions consider performing a transformation to make the distributions more Gaussian, or avoid a least squares





formulation and instead use a method that avoids outliers (e.g. absolute deviations – Trudinger et al., 2007).

- Analyse and correct for biases in the observations, or approximately account for it in the observation error covariance matrix, **R**, using the off-diagonal terms or inflated errors (Chevallier, 2007), or by using the prior model-data RMSE to define the observation uncertainty.

- Investigate potential incompatibilities between your model and data. Take time to understand which physical quantities your data correspond to and whether that is consistent with the description of the equivalent variable in the model. As for the previous point, one way of attempting to account for unknown inconsistencies between the model and data is to set the observation uncertainty, **R**, the prior RMSE between the model and the data.

- Evaluate the impact on other model variables with independent observations, and if the optimisation degrades the fit compared to the prior, investigate the reasons behind the inconsistency and address them as above.

- Assess the non-linearity of your model (multiple first guess tests can help in this regard), and if strongly so, avoid a least squares formulation of the cost function or use global search algorithms for the optimisation – although at the resolution of typical LSM simulations ($\geq$0.5x0.5°) this will likely only be computationally feasible at site or multi-site scale.

- Prior information is key in a Bayesian framework. Effort should be put into better constraining the prior parameter bounds of all parameters based on literature wherever possible.

- Conduct preliminary sensitivity analyses to determine which parameters should be constrained by each data stream.

- Set up experiments with synthetic data, as in this study, to understand the constraints posed by the different data streams you will include in the experiment.

- If technical constraints require a step-wise approach is used it is preferable (from a mathematical standpoint) to propagate the full parameter error covariance matrix



between each step, if it can be calculated, and carefully consider the order of the
assimilation of data streams (a synthetic experiment will aid in this regard).
•   Be aware that a good theoretical reduction in model or parameter uncertainty can
be misleading, as it is not necessarily indicative that the right parameter values
have been found. If this is the case, it could impact predictions made outside the
spatio-temporal window included in the optimisation.
Many of these issues are relevant to any data assimilation study, including those only
using one data stream. However, most are more pertinent when considering more than one
source of data. The impact of bias in the magnitude of satellite-derived FAPAR data has
featured highly in past multiple data stream assimilation studies. Aside from simple
corrections, Quaife et al. (2008) and Zobitz et al. (2014) suggested that LSMs should be
coupled to radiative transfer models to provide a more realistic and mechanistic observation
operator between the quantities simulated by the model and the raw radiance measured by
satellite instruments. This proposition followed the experience gained in the case of
atmospheric models for several decades (Morcrette, 1991).
Other promising directions could also be considered to help constrain the problem of lack
of information in resolving the parameter space, including the use of other ecological and
dynamical "rules" that limit the optimisation (see for example Bloom and Williams, 2015), or
the addition of different timescales of information extracted from the data such as annual
sums (e.g. Keenan et al., 2012). Of course, optimising the parameters of the model will not
account for all the uncertainty in a model. Inaccurate or incomplete process representation is
likely a key factor that may also bias the posterior values retrieved in any optimisation.
Keenan et al. (2012) reflected that despite using multiple different constraints and different
time increments in the cost function, the inter-annual variability and long-term trend of carbon
uptake at Harvard forest FLUXNET site in the USA could not be reproduced without a
temporal variation of the parameters, suggesting a missing process in the model. However, as
this paper shows, the complexities of model-data fusion require that we continue to develop
DA techniques alongside development of LSMs, with the hope of converging upon more
reliable and accurate predictions of the global C budget in the near future. Finally we should
also seek to develop collaborations with researchers in other fields who may have advanced
further in a particular direction. Members of the atmospheric and hydrological modelling





communities, for example, have implemented techniques for inferring the properties of the
prior error covariance matrices, including the mean and variance, but also potential biases,
autocorrelation and heteroscedasticity, by including these terms as "hyper-parameters" within
the inversion (e.g. Michalak et al. 2005; Evin et al., 2014; Renard et al., 2010; Wu et al.
2013;). Of course this extends the parameter space – making the problem harder to solve
unless sufficient prior information is available (Renard et al., 2010), but such avenues are
worth exploring.
## 5  Conclusions
In this study we have attempted to highlight and discuss some of the challenges
associated with using multiple data streams to constrain the parameters of LSMs, with a
particular focus on the carbon cycle. We demonstrated some of the issues using two simple
models constrained with synthetic observations for which the 'true' parameters are known.
We performed a variety of tests in Section 2 to demonstrate the differences between
assimilating each data stream separately, sequentially (in a step-wise approach) and together
in the same assimilation (simultaneous approach). In particular we focused on difficulties that
may arise in the presence of biases or inconsistencies between the data and the model, as well
as non-linearity in the model equations. In Section 3 we discussed the experimental results
with reference to similar difficulties that have been documented in recent C cycle assimilation
studies.
Many of the issues faced are inherent to all optimisation experiments, including those in
which only one data stream is used. It is of upmost importance to determine if the
observations contain biases, and/or if inconsistencies or incompatibilities exist between the
model and the observations, and to correct for this or properly account for this in the error
covariance matrices. Secondly it is crucial to understand the assumptions and limitations
related to the inversion algorithm used. Without these two points being met, there is a greater
risk of obtaining incorrect parameter values, which may not be obvious by examining the
posterior uncertainty and model-data RMSE reduction. Furthermore it is more likely that the
implementation of a step-wise versus simultaneous approach will lead to different results.
This study was not able to examine an exhaustive list of all possible challenges that may
be faced when assimilating multiple data streams, but we hope that this tutorial style paper
will serve as a guide for those wishing to optimise the parameters of LSMs using the variety





of C cycle related observations that are available today. Furthermore we hope that by
increasing awareness about the possible difficulties of model-data integration, we can further
bring the modelling and experimental communities together to work more closely on these
issues.
**Code availability**
The model and inversion code will be made available via the ORCHIDAS website (upon
registration): https://orchidas.lsce.ipsl.fr/multi_data_stream.php.
**Acknowledgements**
We acknowledge the support from the International Space Science Institute (ISSI). This
publication is an outcome of the ISSI's Working Group on "Carbon Cycle Data Assimilation:
How to Consistently Assimilate Multiple Data Streams". N. MacBean was also funded by the
GEOCARBON Project (ENV.2011.4.1.1-1-283080) within the European Union's 7th
Framework Programme for Research and Development. The authors wish to thank colleagues
and collaborators in the atmospheric inversion and C cycle DA communities with whom they
have had numerous past conversations that have led to an improvement in their understanding
of the issues presented here.



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





1  Table 1: The optimisation set-up for both models, including the true parameter values, their

2  range and the observation uncertainty (1 sigma). The parameter uncertainty (1 sigma) was set

3  to 40% of the range for each parameter.

| Model | Parameter value (range) | | | | Observation uncertainty | |
|---|---|---|---|---|---|---|
| Simple carbon | $p_1$ | $p_2$ | $k_1$ | $k_2$ | $s_1$ | $s_2$ |
| model | 1 (0.5,5) | 1 (0.5,5) | 0.2 (0.03,0.9) | 0.1 (0.01,0.12) | 0.5 | 5 |
| Non-linear | $a$ | | $b$ | | $s_1$ | $s_2$ |
| toy model | 1 (0,2) | | 1 (0,2) | | 0.5 | 0.5 |





1    Table 2: List of experiments performed for both models with synthetic data. All parameters

2    are optimised in all cases (therefore in both steps for the step-wise approach).

| Test case | Step 1 | Step 2 | Parameter error covariance terms propagated in step 2? |
|---|---|---|---|
| *Separate* | | | |
| 1a | $s_1$ | - | - |
| 1b | $s_2$ | - | - |
| *Step-wise* | | | |
| 2a | $s_1$ | $s_2$ | yes |
| 2b | $s_1$ | $s_2$ | no |
| 2c | $s_2$ | $s_1$ | yes |
| 2d | $s_2$ | $s_1$ | no |
| *Simultaneous* | | | |
| 3a | $s_1$ and $s_2$ | - | - |
| 3b | $s_1$ and only 1 obs for $s_2$ | - | - |





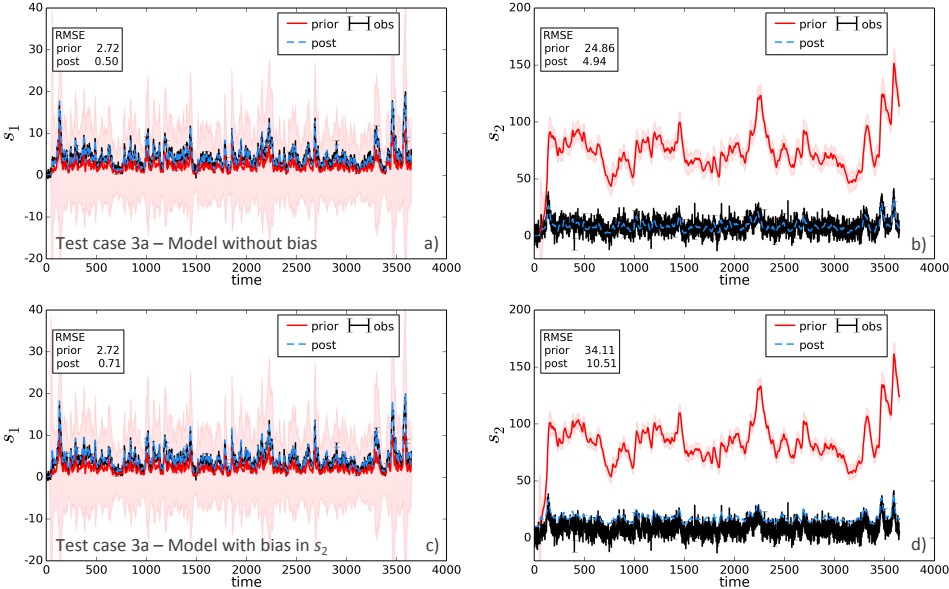

Figure 1: Prior and posterior model simulations compared to the synthetic observations for the
simple carbon model for test case 3a for a) $s_1$ and b) $s_2$ simulations without any model bias,
and c and d) with bias in the simulated $s_2$ variable. The coloured error band on the prior and
posterior represents the propagated parameter uncertainty (1 sigma) on the model state
variables (in the equivalent colour as the mean curve). This is mostly visible for the prior
model simulation (pink band) as there is a high reduction in model uncertainty reduction as a
result of the assimilation.

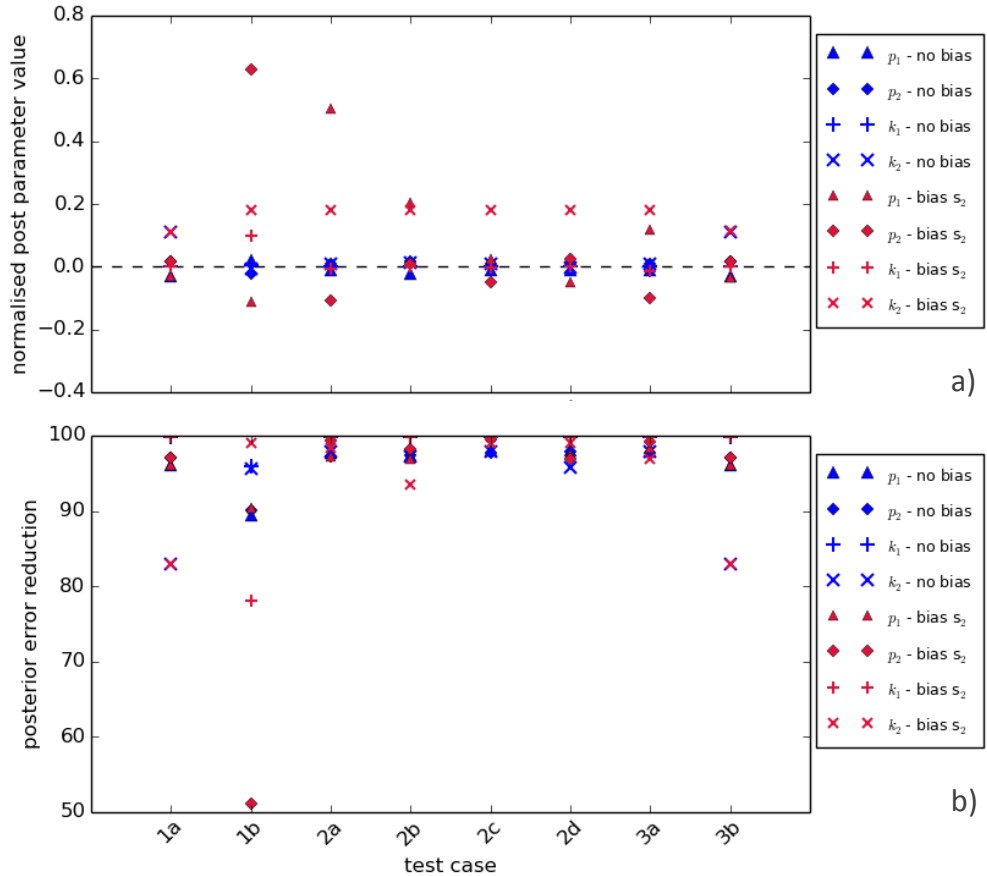

Figure 2: a) Normalised posterior parameter values and b) posterior parameter error reduction
for all parameters of the simple carbon model for each test case, and for both the simulations
with no bias (blue) and simulations with a bias in the $s_2$ variable that was not accounted for in
the inversion (red). In a) parameters values were normalised to account for differences in the
magnitude of the different parameters and their range, thus it is a measure of the distance
from the true value as a fraction of the range and is calculated as: (posterior value – true value
/ max parameter value – minimum parameter value). The closer the value to the zero dashed
line represents a better match to the "true" parameter value. To give an indication of the
optimisation performance, the following are the normalised first guess parameter values for
this particular example test (compare with posterior values in Fig. 2a): $p_1$ 0.09, $p_2$ 0.29, $k_1$ 0.1,
$k_2$ 0.15.





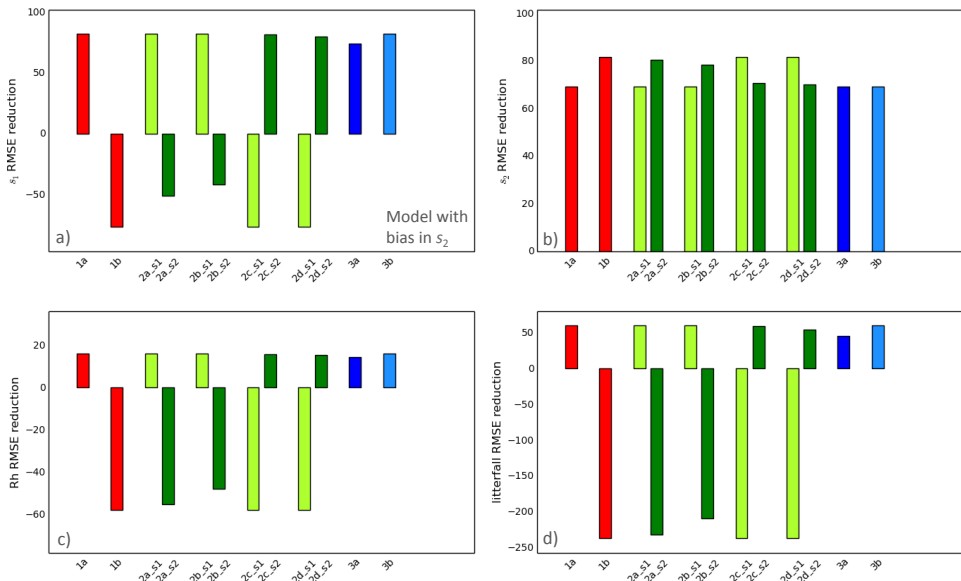

Figure 3: Reduction in RMSE for all test cases for simulations with a bias in the $s_2$ variable: a)
$s_1$, b) $s_2$, c) litterfall and d) heterotrophic respiration (Rh). For the step-wise cases (2a, b, c and
d) the reduction after both step 1 and step 2 are shown in light and dark green respectively,
and are denoted in the x-axis labels with '_s1' for step 1 and '_s2' for step 2. The reduction
(in %) is calculated as $1 - (\mathrm{RMSE}_{post} / \mathrm{RMSE}_{prior})$.



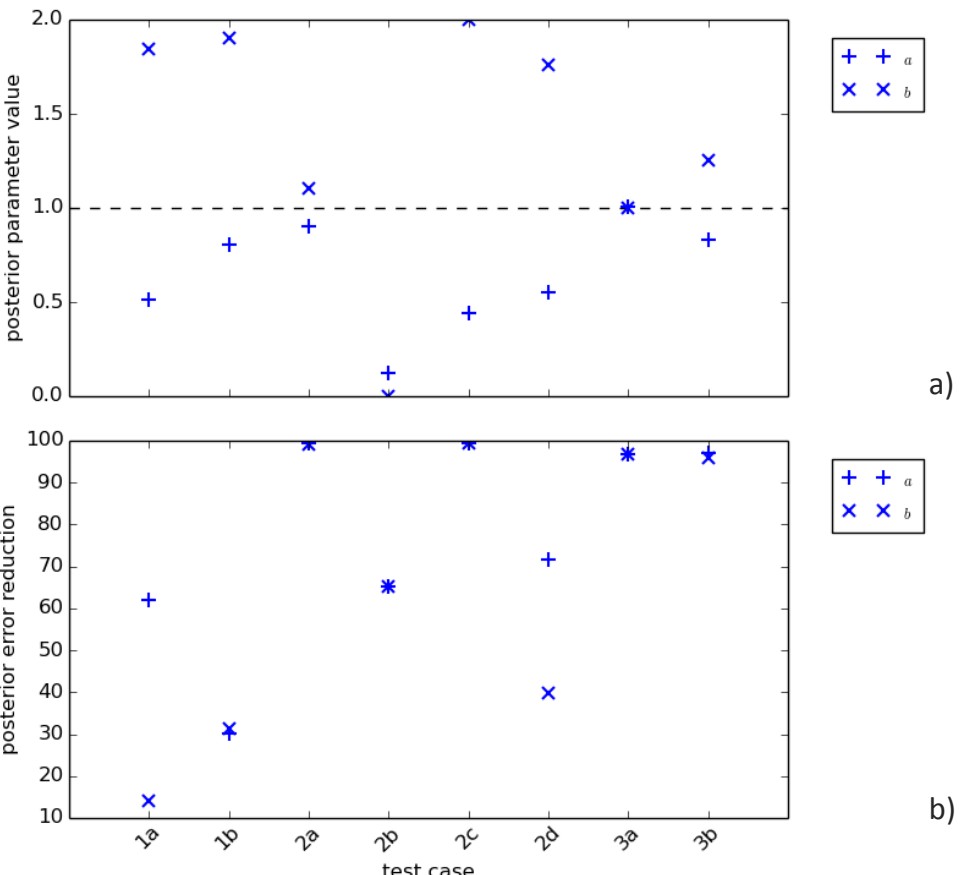

Figure 4: Posterior parameter values of both the non-linear toy model *a* and *b* parameters for each test case for the simulations with no model bias. The y-axis range corresponds to the parameter bounds and the dashed horizontal line represents the "true" known value of both parameters. To give an indication of the optimisation performance, the following are the first guess parameter values for this particular example test (compare with posterior values in Fig. 4a): *a* 0.87, *b* 1.98. b) Posterior uncertainty reduction for both parameters for all test cases.



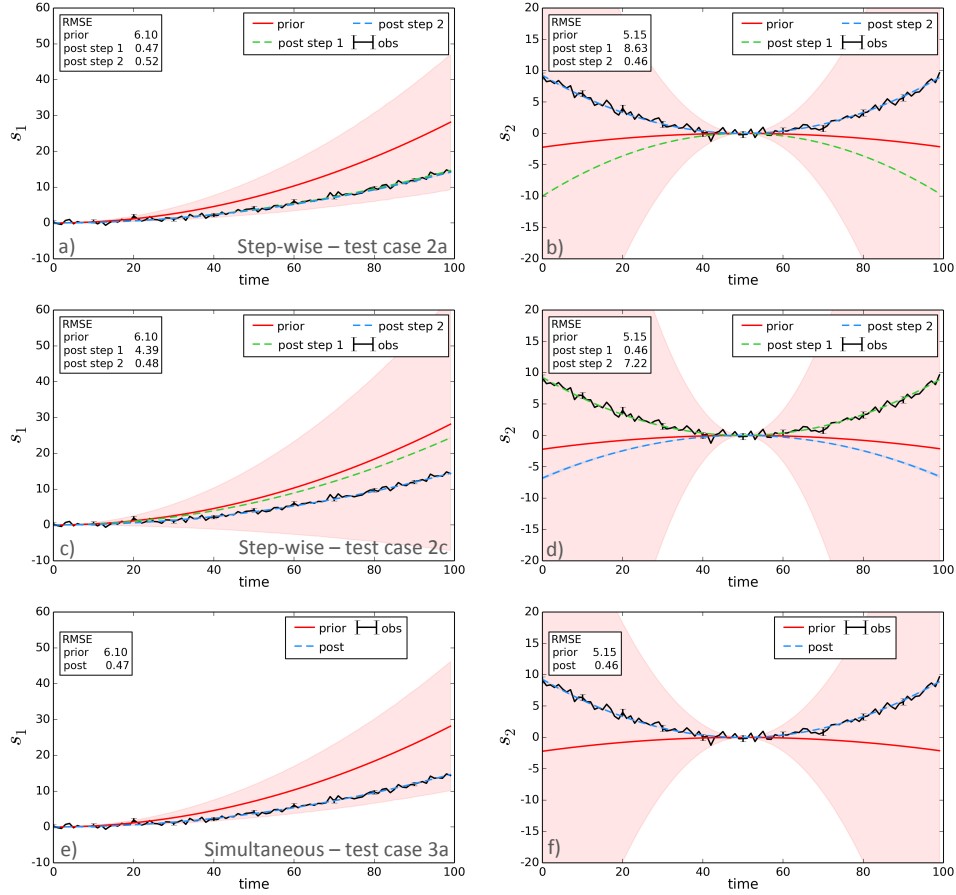

Figure 5: Prior and posterior model simulations compared to the synthetic observations for the
non-linear toy model (with no bias) for both the $s_1$ (left column) and $s_2$ (right column)
variables for a) and b) test case 2a (1$^{st}$ row) – step-wise approach with $s_1$ observations
assimilated in the first step, followed by the $s_2$ observations in the second step; c) and d) test
case 2c (2$^{nd}$ row) – step-wise approach with $s_2$ observations assimilated in the first step,
followed by $s_1$ observations in the second step; and e) and f) test case 3a (3$^{rd}$ row) – the
simultaneous case in which both data streams were included. For both step-wise examples $\mathbf{A}_1$
was propagated between the 1$^{st}$ and 2$^{nd}$ steps. The coloured error band on the prior and
posterior represents the propagated parameter uncertainty (1 sigma) on the model state
variables (in the equivalent colour as the mean curve). This is mostly visible for the prior



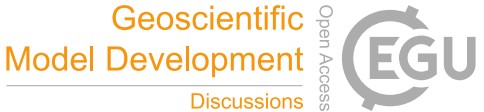

1    model simulation (pink band) as there is a high reduction in model uncertainty reduction as a

2    result of the assimilation.



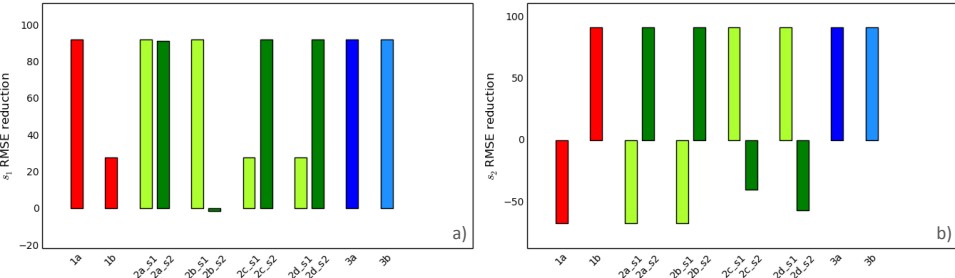

Figure 6: Reduction in RMSE for all test cases for both a) $s_1$ and b) $s_2$ variables for the non-
linear toy model simulations with no model bias. For the step-wise cases (2a, b, c and d) the
reduction after both step 1 and step 2 are shown in light and dark green respectively, and are
denoted in the x-axis labels with '_s1' for step 1 and '_s2' for step 2. The reduction (in %) is
calculated as $1 - (\text{RMSE}_{\text{prior}} / \text{RMSE}_{\text{post}})$.

