# Peer review of "Consistent assimilation of multiple data streams in a"

_Geoscientific Model Development, 2016_

## Referee Comment (RC1) · Anonymous Referee #1 · 8 Apr 2016

General comments:

The paper addresses the question of the assimilation of multiple data streams to estimate model parameters and initial conditions together with their uncertainty using variational method for simple C cycle models using synthetic observations.

The paper is organized around two parts. A first part presenting a variational data assimilation (DA) experiment for a simple, yet non trivial, quasi-linear model for the carbon cycle, and a non-linear toy model using multiple (2) data streams. The DA method, 4DVAR, and the experimental setup are succinctly but clearly described. The results of the experiments are extensively exposed, but - while VAR provides (via adjoint techniques) a set of tools to analyse the DA problem - not explained. The second part is devoted to a rather long but factual literature review of the studies using multiple

data streams to constrain LMS in general and their carbon component in particular.

While the paper illustrates some of the challenges of the model-data fusion problem, it does not describe any new idea, concept or tool, and thus does not represent a sufficiently substantial advance in modelling science. The advices presented at the end of the paper describes the golden rules for any DA experiment and the manuscript would benefit from a strict application of these advices. As it stands the paper only reproduces what other studies have done: performing the assimilation of multiple data streams following different scenarii.

I would recommend to shorten the literature review and to insert it before the experimental study and to perform a thorough analysis including sensitivity analysis, nonlinearity issues, conditioning of the problem, information content. Due to its apparent complexity and because of "the burden" of coding and maintaining an adjoint, VAR is not the most popular method within this field, however it offers a frame- work where diagnostic and prognostic tools can be clearly (and sometimes analytically) defined, the capabilities of VAR deserve to be fully exploited in the scope of this paper.

Specific comments:

- Page 2: "Observations allow us to understand the system up until the present day, but they cannot tell us about the future (...). They also cannot distinguish between the complex interactions that may occur between different processes". I strongly disagree with this statement, observations do carry information about the future through the deterministic processes that, we believe, govern our world.

- Pages 5-6, lines 12-18: I found the input/output terminology on page 5, line 17-19, a bit misleading. A brief summary of dynamics of the model as described in the work of Raupack 2007 could be useful. The models and the dynamic variables they describe try to encompass different time scales from diurnal to potentially much longer time scales, and the variables themselves are likely to differ by several order of magnitudes. A discussion about the implication of the different typical scales could enlighten some

of the challenges. In the description of the experiments details concerning the time step size, observation window and observation frequency could be useful.

- Page 6, line 28: "including measurement and model errors", how to include model error without a weak constraint formulation?

- Page 7, line 6: "strong linear dependence of the model to the parameters", 4DVAR is the perfect framework where this issue should and could be investigated as advised in the section "advice for LMS modellers".

- Page 7, line 2: statement page 7 line 2 requires the model/observation operator to be linear as is discussed on page 17 line 20-31.

- Page 9 : concerning the experiment where only one observation for s2 is considered, worth mentioning that it corresponds (does it?) to the situation where only one estimation, say for soil C stock, is available. In this case is it used as a prior for s2 or as an observation later in the time window thus allowing the model to create correlation with other variables and parameters?

- Page 11, lines 14-17: discussion about "good or moderate reduction in RMSE for variables not included in any assimilation (...)" why is the reduction so poor for this flux? can this be expected from a model sensitivity analysis.

- Page 18, lines 16-19: "Rather if the model sensitivity to the parameters is very non-linear, multiple combinations of parameter values may exist that result in a similar reduction of the cost function (multiple minima), but provide a different fit to each data stream". This is exactly a crucial aspect that the paper should focus on, simplified and toy models are meant for this.

- Page 18, line 20: information content not defined, and more generally the expression "enough information" appear twice in the text but never made explicit.

- Page 18, lines 23-29: how to find the "troublemaker" and "peacemaker"?

[Figure]

- Page 26, lines 16-17: biases and inconsistencies, and other problematic features, could be addressed prior to optimisation in the context of the linearisation of the model.

- Page 29, lines 25-26: "it is crucial to understand the assumptions and limitations related to the inversion algorithm used" yet I feel that the paper did not provide the analysis, though possible with VAR, that would have helped understanding these assumptions and limitations in the case of the "simple" models presented here.

Technical corrections: - On page 1 line 25: "data stream" instead of "data steam". - On page 3 line 30: "matrices" instead of "matrixes". - In Table 1 : for the non-linear toy model the observation uncertainty for s2 is set to 0.5 whereas it is set to 5 for the simple carbon model, shouldn't it be 5 instead of 0,5?

———————————————————

---

## Referee Comment (RC2) · Anonymous Referee #2 · 11 Apr 2016

This manuscript examines aspects of assimilating multiple data streams into carbon cycle models, includes discussion of the preceding literature and makes recommendations for the carbon cycle data assimilation (DA) community as to best practice when performing DA experiments. A real strength of this paper lies in the clarity of the description of the Data Assimilation problem.

Overall the work presented is well written, appears technically sound and should be easily reproducible. However the value of the individual parts of the manuscript feel somewhat limited, and as a whole I am not convinced they combine to make a complete piece of work. Although I don't doubt that setting up the DA system itself was technically complex, the experiments performed with it are rather limited in scope. My feeling is that it would have been easy to explore some further aspects of the carbon cycle DA problem and make the resulting manuscript much stronger with relatively little extra

work.

The "advice for land surface modellers" in section 4 is a good concept but could be better organised. For example the points "conduct preliminary..." and "set up experiments..." are very related. I think the list should be tidied up - perhaps broken into different sections, for example "understanding errors", "preliminary analyses" and so on. Each of these sections can then contain the smaller points.

The literature review section is reasonable but does not go into some of the preceding work in sufficient depth. In particular there are two studies I can think of that also look at carbon cycle DA problems with simple models that should have been dealt with in more detail. The Optic paper by Trudinger et al. (2007) is referenced, but a discussion of what experiments were performed and what they authors found is lacking. I think this is an important oversight given that this manuscript uses the same model. The Reflex paper by Fox et al. (2009) which looks at parameter estimation using a variety of DA techniques using a simple model and synthetic data isn't referenced. Furthermore the ordering of the manuscript feels a bit backward. One would normally expect the literature review to come prior to the experimental component and to set up the rationale for the experiments that follow.

I have the following major recommendations to make the manuscript publishable:

1) The experiments performed with the model need to be broader. There are several issues brought up later in the manuscript which could be easily examined. For example some simple experiments looking at populating the off-diagonal elements of the R matrix to set correlation between observations of S1 and S2 would seem to be an easy thing to do. I would be happy to see any sensible additional experiment though.

2) The literature review should be moved before the experimental section and modified so that it builds the rationale for performing the specific experiments undertaken. It should include greater discussion of the papers mentioned above. There are also classic problems in data assimilation which have not been well investigated in the car-

bon cycle to date such as localisation and errors or representativity and these have not been mentioned. They should be added into the discussion.

3) The "advice" list needs to be re-written to provide a bit more order. See comments above.

4) On page 11 at line 27 there is a statement suggesting that the data streams of s1 and s2 contain enough information to retrieve all the parameters individually for the quasi-linear model. This to me seems to be a flaw in the experimental design. Some of the conclusions from this part of the paper revolve around the linearity of the model, e.g. that differences between the step-wise and simultaneous experiments are minimal because of this. However given that the model is such that either set of observations can be used to determine both parameters it is not possible to say definitively that is the models linearity which is responsible for this. My hunch is that the authors are correct, but what would happen with a more complex linear model where not all parameters are observable from either one data stream? The only way to demonstrate this is by introducing a new model - which I do not recommend - however I think it is vital that the authors are clear about what can or cannot be deduced from these experiments.

I have the following minor comments:

1) The first paragraph of page 4 makes a lot of statements that are not referenced. It would be helpful to the reader who wanted to follow up on some of these aspects to provide references.

2) On page 5 I felt a bit more information was required about the model. How is the value of the functions F(t) being evaluated (possibly I have just misunderstood what is going on - so maybe just some clarification is needed).

3) Page 23, line 4, I am not sure what is meant by orthogonal here. Given that S1 and S2 are interdependent on each other in the quasi-linear model the observations of them (assuming the model is correct, which it is in these synthetic experiments) cannot
be not orthogonal. Perhaps the word "additional" would be better used here? Either that or I think the choice of "orthogonal" needs to be justified.

Typographic and small errors:

P03L10: step -> steps P03L19: one -> only P12L11: uniform -> constant (?) P13L11: than -> as P15L4-L11: this sentence needs to be broken up for clarity. P29L05: 2013.). -> 2013).

F2a: y-label should read "posterior" instead of "post"? F2b: y label should contain "%". F3caption: Equation should be 1-(RMSE_post/RMSE_prior)x100 F4b: as F2b

References:

Trudinger, Cathy M., et al. "OptIC project: An intercomparison of optimization techniques for parameter estimation in terrestrial biogeochemical models." Journal of Geophysical Research: Biogeosciences 112.G2 (2007).

Fox, Andrew, et al. "The REFLEX project: comparing different algorithms and implementations for the inversion of a terrestrial ecosystem model against eddy covariance data." Agricultural and Forest Meteorology 149.10 (2009): 1597-1615.

---

## Author Comment (AC1) · 27 Jun 2016

Response to Interactive comment on "Consistent assimilation of multiple data streams in a carbon cycle data assimilation system" by Natasha MacBean et al. Anonymous Referee #1

General comments: The paper addresses the question of the assimilation of multiple data streams to es- timate model parameters and initial conditions together with their uncertainty using variational method for simple C cycle models using synthetic obser- vations. The paper is organized around two parts. A first part presenting a variational data assimilation (DA) experiment for a simple, yet non trivial, quasi-linear model for the carbon cycle, and a non-linear toy model using multiple (2) data streams. The DA

method, 4DVAR, and the experimental setup are succinctly but clearly described. The results of the experiments are extensively exposed, but - while VAR provides (via adjoint techniques) a set of tools to analyse the DA problem - not explained. The second part is devoted to a rather long but factual literature review of the studies using multiple streams to constrain LMS in general and their carbon component in particular. While the paper illustrates some of the challenges of the model-data fusion problem, it does not describe any new idea, concept or tool, and thus does not represent a sufficiently substantial advance in modelling science. The advices presented at the end of the paper describes the golden rules for any DA experiment and the manuscript would benefit from a strict application of these advices. As it stands the paper only reproduces what other studies have done: performing the assimilation of multiple data streams following different scenarii.

» RESPONSE We thank the reviewer for their comments and detailed review.

i) Firstly we would like to address the assertion that the paper does not describe any new idea, concept or tool, and therefore does not represent a sufficiently substantial advance in modelling science. While indeed there is no new idea, concept or tool, this is a model experiment description paper to elucidate the concepts and problems for multiple data stream assimilation, particularly in reference to large-scale, complex land surface models (LSMs) that are included in earth system models (ESMs), and that may have to rely on a variational data assimilation method (more on this below in the comments about the VAR framework). To our knowledge this is the first paper that has brought together a investigation of the issues surrounding multiple data stream assimilation for LSMs from a methodological point of view. Many papers have used multiple data streams, as described in the literature review section, but very few (or none – to our knowledge) have highlighted the impacts and challenges around this topic. With the growing increase in the number and length of data streams, we tend to think that adding more data streams will definitely be beneficial for optimising a model. This paper aims to show that whilst that may be true, it is not the magic "black box"

that researchers may hope for, and many factors must be considered when carrying out this type of assimilation experiment, particularly when assimilating several different data streams of a different nature (i.e. flux and stocks, satellite data) or density (number of measurements). We agree however that the structure of the major sections of the paper (with the literature review after the experimental section) does not help to illustrate that this is indeed an open issue in land surface modelling and that we try to solve this by looking at two simple but representative models (more on this below).

We tried to provide such a justification for the work in the introduction with paragraph starting:

P3 line 22: "Increasingly, researchers are attempting to bring these sources of information together to constrain different parts of a model at different spatio-temporal scales within a multiple data stream assimilation framework (e.g. Richardson et al., 2010; Keenan et al., 2012; Kaminski et al., 2012; Forkel et al., 2014; Bacour et al., 2015). However, whilst the potential benefit of adding in extra data streams to constrain the C cycle of LSMs is clear, multiple data stream assimilation is not as simple as it may seem"

This paragraph goes on to explain the issues further, but as mentioned we agree that we could make this more clear, particularly its relevance to optimisation of large-scale, complex LSMs, therefore we have added the following sentence after the sentence above, which we hope will strengthen the justification for this work:

"This is particularly true when considering a regional-to-global scale multiple data stream, multiple site optimisation of a complex LSM that contains many parameters, and which typically takes on the order of minutes to an hour to run a one year simulation."

ii) Secondly, we are not completely sure what the reviewer wants to say with this comment: "The results of the experiments are extensively exposed, but - while VAR provides (via adjoint techniques) a set of tools to analyse the DA problem - not explained".

We have tried to explain the results we see carefully. Please could the reviewer provide further details on how they would like to see the results explained more? However, we would also like to emphasise and to remind the reviewer that the paper was not intended to explore the strength of variational methods, or the data assimilation set-up as a whole, but to demonstrate the issues with multiple data stream assimilation (e.g. issues of non-linearity when considering a step-wise or simulteanous simulation – we have given more details on this below). Therefore we do not want to add too many prior or posterior diagnostic tests as we already feel the paper is long enough. Furthermore, reviewer 2 has suggested adding additional tests that are more related to the specific issue of multiple data streams (e.g. the impact of having correlated errors between the observations). We are going to do the test requested by reviewer 2 as we feel it more directly fits in with the aims of the paper, whereas although diagnostic tests are a nice addition they may make the paper too long.

Having said the above, we do feel that the aim of the paper, to specifically explore the issues of multiple data stream assimilation, has not been brought out enough in the text. In paricular, as the reviewer notes, the advice presented at the end of the paper are golden rules for any DA experiment, and thus they are too general for the stated aims of the paper. We agree this muddies the manuscript focus. Therefore we can understand that the reviewer was misled by this, which led to their comment about using "VAR as a set of tools to analyse the DA problem (in general)". We have therefore changed the text as stated in point i) above, and in addition we have changed the advice to land surface modellers section (4) to remove any points that are general to a DA experiment, so as not to confuse the reader, and to reflect more on the results (see comment below). We have put the revised text for this section at the bottom of this review.

Lastly we would like to insist that this is not a simple replication of what other studies have done. We believe our work is different in the following ways: - No study (to our knowledge) has specifically investigated the difference between step-wise and

simultaneous optimisation for multiple data streams, which is important in the context of large-scale LSMs using variational methods due to the heavy computational burden, as a step-wise optimisation may be preferable (as discussed from P3 line 27 onwards in the introduction). - Similarly, no study has investigated the impact of bias/incompatibility/inconsistency in the observations or model on the assimilation results, again which is particularly important in the context of large-scale LSMs using variational methods due to the heavy computational burden - No study has used synthetic observations and simple models to demonstrate these issues of multiple data stream assimilation when using complex large-scale LSMs that necessarily require the use of a variational schemes for land surface modelers - Whilst some studies have noted that using one data stream can degrade the fit to the another, only a couple have fully explored why this is the case. In this study we explore issue this with the aid of synthetic observations and simple toy models, making it much easier to see the potential negative impact of such a situation.

We have tried to bring these points out more in the introduction, by putting the literature review before the experimental section as advised, by building the case for the following experiments with the simple models more within the literature review, by altering the advice section to be more focused on multiple data stream assimilation and finally by adding an extra introductory paragraph at the start of the experimental results section (see response *** below). »

I would recommend to shorten the literature review and to insert it before the experimental study and to perform a thorough analysis including sensitivity analysis, nonlinearity issues, conditioning of the problem, information content.

» RESPONSE We agree with the reviewer about the order of the sections, and have put the literature review before the experimental study as discussed above. We have also shortened it by taking out any references to the experimental results (P21 lines 22 to 27 – which was added to the advice section as the literature review is now before the results) and P22 lines 1 to 8 has also been added to the advice section as these

sentences accompanied P21 lines 22 to 27). In addition, as recommended by the reviewer, we have deleted a few other sections in both the literature review that were superfluous (e.g. P21 lines 14-15 and lines 27 to 32, P22 lines 9 to 14) as well as sentences/bullet points in the advice section that were more related to general issues in data assimilation, i.e. even just with one data stream (all noted in the revised document, for example P28 line 21 to 30). However, reviewer 2 asked for additional discussion of two key papers that had not yet been discussed, so we have added this in to the literature review.

We have not included a preliminary sensitivity analysis for the following reason. Normally we would perform a sensitivity analysis either a) when optimising a more complex model with many different processes and parameters with data streams that only correspond to one part of the model, or b) if the computational time is long enough that excluding non-sensitive parameters becomes important for computational efficiency. Neither is the case here because the toy models in this paper have so few parameters (2 and 4), all of which we want to optimise because the model variables are sensitive to all parameters. All observations constain all parameters. As we have seen from a preliminary analysis of the Jacobian, all observations constrain all parameters, the relative constraint from each observation may change within the time window and will depend on the specific DA set-up including the trajectory within parameter space, the specific noise realisations (added random noise) etc. Investigating all this is beyond the scope of this paper because these are general DA issues, not pertinent to a multiple data stream assimilation paper. Therefore we do not wish to include a specific sensitivity analysis in this paper. However, to explain this to the reader we have added the following sentence in (now) Section 3.1.5 (P9 line 11 in the original manuscript): "We have not performed a prior sensitivity analysis to decide to which parameters are important to include in the optimisation, as the model variables are sensitive to all of the (small set of) parameters. However, in the case of a more complex, large-scale LSM it is advisable to carry out such an analysis, particularly given the computational burden of optimising many parameters."

We agree that showing the non-linearity (or lack thereof, for the first model) of the two models would be beneficial for the reader. Therefore we have plotted 3D plots with the pairs of parameters on the x-axes and the cost function on the y-axis to show this. We have put this figure in the supplementary material and refer to it when describing the non-linear toy model in Section 3.1.2.

Conditioning of the problem is an important issue in general for DA, but as mentioned before we are not aiming to provide a general demonstration of all options in a DA set-up. Here we did not condition the problem (normally done by scaling the parameter values to a certain range or normalising by the parameter error covariance) as this was not necessary with the small set of paramters. Analysing the conditioning of the problem and information content (or observation influence) are interesting and useful aspects of a DA set-up, but we would do not want to specifically address this in this paper with different tests for the reasons mentioned above – that it is not the focus of a multiple data stream assimilation paper (therefore will dilute the message) and that these types of additional analyses will make the paper too long, especially given that we will add tests related to the correlated errors between data streams. However given the reviewer's comment below ("Page 18, line 20") on how we define to information content, we have further clarified in the text. »

Due to its apparent complexity and because of "the burden" of coding and maintaining an adjoint, VAR is not the most popular method within this field, however it offers a frame- work where diagnostic and prognostic tools can be clearly (and sometimes analytically) defined, the capabilities of VAR deserve to be fully exploited in the scope of this paper.

» RESPONSE Side note: Variational methods (VAR) refers to the method of adjusting the intial state of a system via assimilating over an assimilation time window, as opposed to sequential methods which update the system at the time of the analysis. IN this context, using derivative-based methods (where the tangent linear model or adjoint is needed) or global search methods could both be used in the context of a variational

scheme. In this paper, we were careful to distinguish between methods, derivative (gradient)-based) and global search. But we see the reviewer is using VAR as it is commonly used, referring to derivative methods for minimising the cost function, which is the method we use in this study.

i) We might agree with the reviewer that "VAR is not the most popular method within this field" when all optimisation studies are taken into account, particularly for site-scale studies. However, VAR is the most popular (or only – to our knowledge) method used when optimising land surface models with multi-site datasets at regional to global scales, because of the computational load of running these larger-scale more complex models. Global search methods simply take too long when doing a large-scale, multi-site optimisation with a land surface model (although new approaches in ensemble methods and updated computational resources will hopefully allieviate this problem). In the literature review (now Section 2), we detail the global scale studies with LSMs that have been carried to this point – to our knowledge - (although we have added a reference to a paper that has been published since our submission – Raoult et al., 2016). All of them use a variational (derivative-based) scheme. It is for this reason that we have chosen to demonstrate these issues using a variational (we refer to this in the paper as a derivative-based for the reason given above) algorithm instead of a global search method (such as the genetic algorithm or MCMC methods). We specifically wanted to show land surface modelers the implications of using variational methods, because whilst many site-scale optimisations have been performed, we are increasingly (and necessarily) moving towards larger scale optimisations with multiple data streams, and in these cases it will likely that they have to use a variational scheme. This is also the reason we have chosen to do the tests with the non-linear model (which is more representative of a more complex LSM), because as a result they may (unknowingly) violate some of the assumptions associated with variational methods given their complexity and possible non-linearities. But we agree that the reasoning for using variational (derivative-based) methods and for doing the non-linear test was perhaps not explained clearly enough, and therefore we have added the following para-

graph to the beginning of the experimental section (now Section 3) which now comes after the literature review section (2):

*** "The three sub-sections in Section 2 highlight examples within a carbon cycle modeling context of the three main challenges faced when performing a multiple data stream assimilation, namely, i) the possible negative influence of including additional data streams into an optimization on other model variables; ii) the impact of bias in the observations, missing model processes or incompatibility between the observations and with the model, and iii) the difference between a step-wise and simultaneous optimization if the assumptions of the inversion algorithm are violated, which is more likely to be the case with non-linear models when using derivative-based algorithms and least-squares formulation of the cost function. The latter point is important because derivative methods (compared to global search) are the only viable option for large-scale, complex LSMs given the time taken to run a simulation."

This (and the rest of the paragraph inclded here, which links the sections in the review to the experimental sections) also answers reviewer's 2 request to build the case for the experiments in the literature review and to link the literature review to the experimental section.

ii) We do not understand the last part of the reviewer's comment above that "VAR specifically offers a framework where diagnostic and prognostic tools can be clearly defined". This is the true for any Bayesian DA framework, including those that use global search methods. In that sense we do not know what the reviewer is referring to when he states that "particular capabilities of VAR (itself) should be exploited in this paper". But we take it to mean that we have not explored included all diagnostic tools that are available within a Bayesian DA framework. As we have discussed above, we think this reflects the reviewer's misunderstanding of the aim of this paper. The assimilation methodology (including exploiting the capabilities of VAR) is not (or at least should not be) a relevant issue for the message of the paper, and therefore we should not include many extra diagnostic tests. Though as discussed we take responsibility
for the misunderstanding over the focus of the paper, and admit that the focus (issues of multiple data stream assimilation, particularly in relation to LSMs) was not defined clearly enough and we have taken steps to address this (see above). »

Specific comments: - Page 2: "Observations allow us to understand the system up until the present day, but they cannot tell us about the future (...). They also cannot distinguish between the complex interactions that may occur between different processes". I strongly disagree with this statement, observations do carry information about the future through the deterministic processes that, we believe, govern our world.

» RESPONSE We agree with the reviewer – we did not intend to make this point so strongly. Therefore we have changed this sentence to: "Observations allow us to understand the system up until the present day and provide inference about how ecosystems may respond to future change. However, their use in estimating model state variables and boundary conditions has limited use beyond diagnostic purposes, and they can be limited in their spatial coverage. They also do not contain all the information we may need to distinguish between the complex interactions that may occur between many different processes" »

- Pages 5-6, lines 12-18: I found the input/output terminology on page 5, line 17-19, a bit misleading. A brief summary of dynamics of the model as described in the work of Raupack 2007 could be useful. The models and the dynamic variables they describe try to encompass different time scales from diurnal to potentially much longer time scales, and the variables themselves are likely to differ by several order of magnitudes. A discussion about the implication of the different typical scales could enlighten some of the challenges. In the description of the experiments details concerning the time step size, observation window and observation frequency could be useful.

» RESPONSE We agree with the reviewer for the most part and have changed the following in Section 3.1.1 ("Simple carbon model"):

i) "The litterfall is an output of s1 and an input to s2 and is a   fraction of the above-

ground carbon reserve as represented by k1s1." → "The litterfall is an output of s1 (aboveground biomass) and an input to s2 (belowground biomass) and is calculated as a constant fraction of the aboveground carbon reserve, defined by k1s1".

ii) We do not want to repeat the description of the dynamical behaviour of the model as detailed in Raupach (2007) because it has already been described in depth by Raupauch (2007) and for the sake of brevity. But we have changed the words "model behaviour" to "dynamical behaviour of the model" (P5 line 22). We have also changed this sentence to be clearer about what this model : "It is based on two equations that describe the temporal evolution of
two carbon pools, s1 and s2:" to: "It is based on two equations that describe the temporal evolution of two living biomass (carbon) pools, s1 and s2, and the biomass fluxes between these two pools"

iii) And finally we have added a part to the following sentence to detail that the model operates at a daily time step: "It is based on two equations that describe the temporal evolution (on a daily time step) of two living biomass (carbon) pools, s1 and s2, and the biomass fluxes between these two pools".

The reviewer is also right that we have not defined the observation window or frequency. We have chosen to add these details into the section on the generation of synthetic observations (now Section 3.1.5 – Optimisation set-up: parameter values and uncertainty, and generation synthetic observations). Therefore after the first sentence of this section we have added the following:

"We optimised a ten-year time window for the simple carbon model, in order to capture the dynamics of the s1 and s2 pools over a time period compatible with typically available observations. For the non-linear toy model, which did not correspond to physical processes in the terrestrial biosphere, we ran a simulation over a window of 100 integration (steps) of the equations. The observation frequency was daily, corresponding to the time-step of the simple carbon model (a value of 1 for the non-linear toy model), and the observation error ... (as above)" »

- Page 6, line 28: "including measurement and model errors", how to include model error without a weak constraint formulation?

» RESPONSE Weak-constraint formulations allow explicitely reducing the model error by including some of its drivers (e.g., a bias in a prognostic equation) in the control vector, provided we have some knowledge of the statistical properties of these drivers. The latter requirement has dramatically reduced the use of this formulation and therefore we only use the standard formulation where the full model uncertainty is simply represented in the observarion error covariance matrix R. In the field of ecosystem model parameter optimisation, this is the standard. »

- Page 7, line 6: "strong linear dependence of the model to the parameters", 4DVAR is the perfect framework where this issue should and could be investigated as advised in the section "advice for LMS modellers".

» RESPONSE Here the reviewer is referring to the description of the assumptions of using the quasi-Newton algorithm (a derivative method) for finding the minimum of the cost function. So indeed, we have explored this issue of the impact of violating the assumption of "strong linear dependence of the model to the parameters" specifically by including a section where we use this framework to try to optimise a non-linear model. We stated this objective in the description of the Non-linear toy model description (now Section 3.1.2), but we have further emphasised this point by changing this sentence: "In order to illustrate the challenges associated with multiple data stream data assimilation for more complex non-linear models, we defined a simple non-linear toy model based on two equations with two unknown parameters" to "In order to illustrate the challenges associated with multiple data stream data assimilation for more complex non-linear models, especially when using derivative methods, we defined a simple non-linear toy model based on two equations with two unknown parameters".

We have also emphasised the point of this test with the paragraph at the beginning of the experimental section (see *** above).

And we have now added in an introductory sentence at the beginning of (now) Section 3.1.6 (Experiments) which says the following: "The specific objective of the following experiments is to test the impact of a bias in the observations that is not accounted for in the R matrix, and the impact of using derivative methods with non-linear models (as may be necessary with large-scale LSMs), particularly with reference the differences that may arise between step-wise and simultaneous optimisations."

These results of this test about using non-linear models with derivative methods were described in now Section 3.2.3 (Difference between the step-wise and simultaneous approaches in the presence of a non-linear model) and discussed further in Section 3.2.5 (Lessons to be learned when dealing with non-linearity).
 Therefore we strongly feel we have addressed this issue, albeit that we did not emphaise this point enough.

As above however we are confused by the fact that the reviewer says that "4DVAR is the perfect framework where this issue could be investigated", as this issue could be investigated within any DA framework. And again as above we do not think that this should be the focus of this paper, except in a context of multiple data stream assimilation. Accordingly we have re-ordered and deleted many points from the advice section to make it much more specific to multiple data stream assimilation, and not about general DA issues relevant to only one data stream. »

- Page 7, line 2: statement page 7 line 2 requires the model/observation operator to be linear as is discussed on page 17 line 20-31.

» RESPONSE Yes, we have added this point to the end of that sentence, thank you. »

- Page 9 : concerning the experiment where only one observation for s2 is considered, worth mentioning that it corresponds (does it?) to the situation where only one estimation, say for soil C stock, is available. In this case is it used as a prior for s2 or as an observation later in the time window thus allowing the model to create correlation with other variables and parameters?

» RESPONSE Indeed it does correspond to this situation, and in this experiment we have taken the first observation (comparable to optimising the initial condition). We have added this point to the sentence the reviewer is referring to (end of P9): "An additional test was included for the simultaneous assimilation in order to test the impact of having a substantial difference in the number of observations for the data stream included in the optimisation, as may be the case for belowground (e.g. soil) biomass observations in reality. Therefore in test case 3b, only one observation was included for data stream s2." »

- Page 11, lines 14-17: discussion about "good or moderate reduction in RMSE for variables not included in any assimilation (...)" why is the reduction so poor for this flux? can this be expected from a model sensitivity analysis.

» RESPONSE Not necessarily, but we may expect that the fit is not as good for variables not included in the assimilation. We could see from a sensitivity analysis if changing the parameters included in the assimilation would change the model variable (i.e. if the model variable is sensitive to those parameters) but we could not know HOW they would affect it, or if the result of the assimilation is closer to the observations or not. In this study we know that all variables are sensitive to all parameters. This is also a general optimisation issue that may be faced when only one data stream is included. »

- Page 18, lines 16-19: "Rather if the model sensitivity to the parameters is very non-linear, multiple combinations of parameter values may exist that result in a similar reduction of the cost function (multiple minima), but provide a different fit to each data stream". This is exactly a crucial aspect that the paper should focus on, simplified and toy models are meant for this.

» RESPONSE We feel we have investigated this in the paper. In particular we did test multiple first guesses precisely to see if we had an issue with multiple minima, or indeed with parameter equifinality. This is detailed at the beginning of the results section (now Section 3.2) with a figure on the reduction in cost function from all twenty

first guesses in the supplementary material. We did not have a problem with multiple minima, we find that in general the same reduction in the cost function is found (as described). However as already discussed at length above we disagree that the paper should focus on this, given it is a paper about multiple data stream assimilation. This is an issue that could arise with one data stream, therefore although we briefly describe the twenty first guess results at the beginning, in order to demonstrate that we have found the global minimum (or at least close to the global minimum), we do not consider that we should go into more depth on this topic in this paper. We do agree it is a key topic, but would be more appropriate for a general DA tutorial, which is not the purpose of this paper. However, we also agree that given we do not focus on the paper, this section is rather speculative and superfluous, and therefore we have removed it (P18 lines 16 onwards). »

- Page 18, line 20: information content not defined, and more generally the expression "enough information" appear twice in the text but never made explicit.

» RESPONSE The reviewer is right that information content here is not defined and we have not been explicit when saying "enough information". We have changed this to agree with how we refer to information elsewhere (e.g. in the introduction we do make several references to what we mean by information at those point, with the sentence: "These data bring information on different spatial and temporal scales,", with the bullet points following that detailing the temporal and spatial scales each data stream contains) so this is now: "spatio-temporal information content" and "enough spatio-temporal information" [to constrain the parameters].

We also agree that in some other places in the text we have not been explicit as to what we mean by information . We have changed this where the word "information" is not clear in the text. For example we have changed P4 line 11 to be: "information on the error covariance", and P22 lines 22 to 24 to be: "The study of Keenan et al. (2013) was particularly notable in its aim to quantify which data streams provide the most information (in terms of model-data mismatch) and how many data streams are

actually needed to constrain the problem". »

- Page 18, lines 23-29: how to find the "troublemaker" and "peacemaker"?

» RESPONSE Given the re-structuring of the advice section (see bottom of the review), and the addition of some aspects of the literature review to the advice section, given the literature review is now before the experimental section, we see that this whole section runs the risk of repeating what is in the advice section, and the discussion provided at the end of this section, including the words "troublemaker" and "peacemaker" (e.g. P18 lines 16 onwards) is somewhat speculative and superfluous (see comment "Page 18, lines 16-19" above). Therefore given another request to streamline and cut down the paper, we have removed this section, and added the following sections to the advice and perspectives section (Section 4):

"Most optimisation studies with a large-scale LSM use derivative methods based on a least-squares approach, and therefore rely on assumptions of Gaussian probability and linear model sensitivity. However,"

"it may not be possible to find the true global minimum of the cost function and the characterisation of the posterior probability distribution will be incorrect. This is a particular problem if the posterior parameter error covariance matrix is then propagated in a step-wise approach, although these issues are relevant to both step-wise and simultaneous assimilation."

"Note that performing a number of tests starting from different random "first guess" points in parameter space can help to diagnose if the global minimum has been reached, as outlined in Section 2.1.6 and discussed at the beginning of the results (Section 2.2)."

We have put the sentence: "An important finding of the results presented for the non-linear toy model in Section 2.2.3 is that degradation in another data stream is not necessarily the result of a bias or incompatibility between the observations and the

model" at the end of Section (now) 3.2.3 (the non-linear toy model section).

Finally we have removed latter part of the section altogether (P18 lines 16 onwards), given the reason above. However, we have added a sentence into the advice and perspectives section that mentions there are several diagnostic tests that can be used if you want to determine the relative influence or constraint brought about by different data streams (e.g. the observation influence metric and the degrees of freedom of signal). We have also added that we have not investigated these metrics here as they are not useful for such simple models with so few parameters, and therefore were beyond the scope of this paper. »

- Page 26, lines 16-17: biases and inconsistencies, and other problematic features, could be addressed prior to optimisation in the context of the linearisation of the model.

» RESPONSE That is true in an ideal case. But in practice it is rarely done unless there is a very obvious bias or inconsistency between the model and the observations (hence why we try to demonstrate its importance in this paper) because in most cases it is not obvious that there is a bias. For example, clearly in the studies we have reviewed, it has not been possible to see the bias in FAPAR data prior to the optimisation. The assimilation revealed this bias. It is not easy to validate satellite data so it is unclear how these biases may be revealed (for this particular example which we have highlighted). »

- Page 29, lines 25-26: "it is crucial to understand the assumptions and limitations related to the inversion algorithm used" yet I feel that the paper did not provide the analysis, though possible with VAR, that would have helped understanding these assumptions and limitations in the case of the "simple" models presented here.

» RESPONSE We feel we have provided this analysis related to the assumptions of linearity requrired by the inversion algorithm, as described above in the reviewer's comment above that begins with "Page 7, line 6". Again, these issues are general DA issues therefore we were not aiming to do a full exploration of all the assumptions related to

the inversion algorithm, but have highlighted those that are particularly pertinent to multiple data stream assimilation, e.g. the results in Section 3.2.3 (Difference between the step-wise and simultaneous approaches in the presence of a non-linear model), which were discussed further in Section 3.2.5 (Lessons to be learned when dealing with non-linearity). We welcome further suggestions from the reviewer on how we can improve the description of the non-linear model section results, which aim to explore these issues in detail. We have attempted to make the point of these experiments clearer, as described in the above comment (Page 7 line 6). »

Technical corrections: - On page 1 line 25: "data stream" instead of "data steam". - On page 3 line 30: "matrices" instead of "matrixes".

» Corrected.

- In Table 1 : for the non-linear toy model the observation uncertainty for s2 is set to 0.5 whereas it is set to 5 for the simple carbon model, shouldn't it be 5 instead of 0,5?

» RESPONSE No, the s1 and s2 observations are very different entities for the different models. The uncertainty was set as a defined 10% of the mean value over the whole timeseries for each pseudo-observation (derived from multiple first guesses of the model). As the magnitude of the s2 observations is larger for the simple carbon model, the associated uncertainty was larger. However the magnitude of s1 and 2 was about the same for the non-linear toy model, so the uncertainty is the same. However, the reviewer has highlighted that this was not defined in the text, therefore we have added the following sentence in (the new) Section 3.1.5 (
[revised manuscript text omitted]

---

## Author Comment (AC2) · 29 Jun 2016

Response to Interactive comment on "Consistent assimilation of multiple data streams in a carbon cycle data assimilation system" by Natasha MacBean et al. by Anonymous Referee #2

This manuscript examines aspects of assimilating multiple data streams into carbon cycle models, includes discussion of the preceding literature and makes recommendations for the carbon cycle data assimilation (DA) community as to best practice when performing DA experiments. A real strength of this paper lies in the clarity of the description of the Data Assimilation problem.

Overall the work presented is well written, appears technically sound and should be easily reproducible. However the value of the individual parts of the manuscript feel somewhat limited, and as a whole I am not convinced they combine to make a complete piece of work. Although I don't doubt that setting up the DA system itself was technically complex, the experiments performed with it are rather limited in scope. My feeling is that it would have been easy to explore some further aspects of the carbon cycle DA problem and make the resulting manuscript much stronger with relatively little extra work.

» RESPONSE

We thank the reviewer for their clear and constructuve review of our manuscript. We understand all his/her concerns about the different parts combining to make a complete piece of work, and we have tried to address these concerns by following all of their suggestions, as detailed below.

»

The "advice for land surface modellers" in section 4 is a good concept but could be better organised. For example the points "conduct preliminary..." and "set up exper-iments..." are very related. I think the list should be tidied up - perhaps broken into different sections, for example "understanding errors", "preliminary analyses" and so on. Each of these sections can then contain the smaller points.

» RESPONSE

We agree with the reviewer on this. We have re-ordered the advice section accordingly taking into the suggestions above. However, we have also deleted many of the points related to general DA issues (e.g. conduct preliminary sensitivity analyses) as we felt, particularly after reading reviewer 1's comments, that this was confusing the focus of the paper on multiple data stream assimilation. Although many of the issues we raise are indeed general issues related to the assimilation of only one data stream, we tried

hard in the manuscript to show how they affected an assimilation with more than one data stream. However we can now see that some of the points made in advice section were counteractive to this goal and could confuse the reader. We hope that in the process we have also tidied up the list, but given the list is now shorter and (hopefully) more focused, we have not broken the points up into sections. However we would be happy to have sub-sections instead of bullets for each of the points. We have put the new advice section at the bottom of this response.

»

The literature review section is reasonable but does not go into some of the preceding work in sufficient depth. In particular there are two studies I can think of that also look at carbon cycle DA problems with simple models that should have been dealt with in more detail. The Optic paper by Trudinger et al. (2007) is referenced, but a discussion of what experiments were performed and what they authors found is lacking. I think this is an important oversight given that this manuscript uses the same model. The Reflex paper by Fox et al. (2009) which looks at parameter estimation using a variety of DA techniques using a simple model and synthetic data isn't referenced. Furthermore the ordering of the manuscript feels a bit backward. One would normally expect the literature review to come prior to the experimental component and to set up the rationale for the experiments that follow.

» RESPONSE

We agree with the reviewer about the structure of the paper and so have put the literature review before the experimental component and slightly re-ordered it to better fit as an introductory section (see point (2) below). We have addressed the suggestions for additional papers below.

»

I have the following major recommendations to make the manuscript publishable: 1)

The experiments performed with the model need to be broader. There are several issues brought up later in the manuscript which could be easily examined. For example some simple experiments looking at populating the off-diagonal elements of the R matrix to set correlation between observations of S1 and S2 would seem to be an easy thing to do. I would be happy to see any sensible additional experiment though.

» RESPONSE

We agree that we could, or should, have added more experiments. Indeed we thought of such experiments from the outset of this work but ended up not including such experiments for fear the paper was too long or the message too complex. We agree that the most obvious, and hopefully most informative, experiment would be one investigating the impact of having correlated observations and populating (or not) the off-diagonal elements of R. We considered examining temporal autocorrelation, but as we want to focus on the multiple data stream aspects we have just considered the correlation between the two data streams. We have implemented this test, but the results we obtained were not what we expected (little impact). As we think this is the most useful extra experiment to include, we have asked the editors for more time (from the 24th June) to investigate this issue and we will provide a further update to this response within the next month. However we will upload the response to the rest of the comments now so the reviewer has more time to look and reply should they wish. The results from these experiments will be presented in a separate section at the end of the experimental section (now Section 3).

»

2) The literature review should be moved before the experimental section and modified so that it builds the rationale for performing the specific experiments undertaken. It should include greater discussion of the papers mentioned above. There are also classic problems in data assimilation which have not been well investigated in the carbon cycle to date such as localisation and errors or representativity and these have not

been mentioned. They should be added into the discussion.

» RESPONSE

We have moved the literature review before the experimental section and have removed some sections, either those that described the experimental results (P21 lines 22 to 27 and P22 lines 1 to 8 – which have now been included in the "advice and perspectives section 4 – see the end of the response), and we have deleted sections that we felt were superfluous, in order to shorten the length as requested by reviewer 1 (e.g. P21 lines 14-15 and lines 27 to 32, P22 lines 9 to 14). We have added in more refererence to the Trudinger and Fox et al. papers but we have not discussed these in too much detail because we want the emphasis of the literature review to be on multiple data assimilation. In this context the Fox et al paper is perhaps more relevant, so we were wrong not to include it in the original text. It is now included in Section 2.1 – Extra constraint from multiple data streams (P21 line 9 before "Thum et al."). The focus of the Trudinger paper is on testing the assimilation set up more than testing issues related to multiple data stream assimilation, and therefore we have not discussed the paper in too much depth. However, given we do want to emphasise the focus on multiple data stream assimilation (please see the response to reviewer 1 for further comments and changes to the manuscript in this regard), we have expanded the last paragraph in the introduction so it starts with the following: "This tutorial-style paper highlights some of the challenges of multiple data stream optimisation of carbon cycle models discussed above. Note that we do not aim to explore all possible issues related to a DA system, for example the choice of the cost function, minimization algorithm, or the characterization of the prior error distributions; indeed previous studies have investigated such aspects at length (e.g. Fox et al., 2009; Trudinger et al., 2007), therefore we refer the reader to these papers for more information. Section 2 reviews recent carbon cycle multiple data stream assimilation studies with reference to some of the aforementioned challenges. Section 3. . ." We hope that these additions are sufficient? Finally we have moved the following section from the advice to the literature review (end of Section 2.2

– impact of bias) because we felt it was better placed there and gave more context to the discussion on bias in FAPAR data seen in previous studies: "Aside from simple corrections, Quaife et al. (2008) and Zobitz et al. (2014) suggested that LSMs should be coupled to radiative transfer models to provide a more realistic and mechanistic observation operator between the quantities simulated by the model and the raw radiance measured by satellite instruments. This proposition followed the experience gained in the case of atmospheric models for several decades (Morcrette, 1991)."

We have tried to further build the rationale for performing the experiments with the toy models throughout the literature review, which we also hope was partly achieved by cutting out speculative and superfluous sections. To further help link the literature review and the experimental section we have added an introductory paragraph to (the new) Section 3 ("Demonstration with two simple models and synthetic data") that summarises the issues raised in the previous section and introduces the experiments at the same time. Therefore the following paragraph has been inserted before Section 3.1 ("Methods"):

"The three sub-sections in Section 2 highlight examples within a carbon cycle modeling context of the three main challenges faced when performing a multiple data stream assimilation, namely, i) the possible negative influence of including additional data streams into an optimization on other model variables; ii) the impact of bias in the observations, missing model processes or incompatibility between the observations and with the model, and iii) the difference between a step-wise and simultaneous optimization if the assumptions of the inversion algorithm are violated, which is more likely to be the case with non-linear models when using derivative-based algorithms and least-squares formulation of the cost function. The latter point is important because derivative methods (compared to global search) are the only viable option for large-scale, complex LSMs given the time taken to run a simulation. This section aims to demonstrate these challenges using simple toy models and synthetic experiments where the true values of the parameters are known. Most importantly this framework

also allows us to investigate the impact of biases and violation of assumptions related to linearity (as discussed in Section 2.2. and 2.3), which are not always evident with real data and large-scale models. Thus the following sections include a description of the toy models together with the derivation of synthetic observations, the inversion algorithm used to optimise the model parameters and the experiments performed, followed by the results for each test case."

Finally, we have included one sentence in the literature review regarding representativity – if I have understood the suggestion of the reviewer correctly ("The spatial distribution of each data stream is also important, especially for heterogeneous landscapes (Barrett et al., 2005; Alton, 2013)") but we did not discuss this or the localisation problem further as we would like to keep the focus on multiple data stream assimilation and not general DA issues. Indeed, we have modified and added certain sentences throughout to try to reinforce this main focus of the paper (see response to reviewer 1).

»

3) The "advice" list needs to be re-written to provide a bit more order. See comments above.

» We agree and have done this – please see the response above and the new text at the bottom.

4) On page 11 at line 27 there is a statement suggesting that the data streams of s1 and s2 contain enough information to retrieve all the parameters individually for the quasi-linear model. This to me seems to be a flaw in the experimental design. Some of the conclusions from this part of the paper revolve around the linearity of the model, e.g. that differences between the step-wise and simultaneous experiments are minimal because of this. However given that the model is such that either set of observations can be used to determine both parameters it is not possible to say definitively that is the models linearity which is responsible for this. My hunch is that the authors are correct, but what would happen with a more complex linear model where not all parameters

are observable from either one data stream? The only way to demonstrate this is by introducing a new model - which I do not recommend - however I think it is vital that the authors are clear about what can or cannot be deduced from these experiments.

» RESPONSE

We thank the reviewer for pointing out the lack of clarity here. Indeed it is not possible to say definitively that it is related to the model linearity and we also feel fairly sure that if we had a more complex linear model that not all parameters would be observable from one data stream. We have therefore clarified this in the text by changing "under this assimilation set-up" to "under this assimilation set-up with this model", and with the sentence at the end of the section: "However, we cannot definitively say whether this is due to the simplicity or relative linearity of the model – it is possible that observations of variables in more complex linear model would not be able to retrieve the true values of all parameters."

»

I have the following minor comments: 1) The first paragraph of page 4 makes a lot of statements that are not referenced. It would be helpful to the reader who wanted to follow up on some of these aspects to provide references.

» RESPONSE

Thank you for pointing this out. We have tried to provide some references. This paragraph now reads: "Mathematically, the optimal approach is the simultaneous, but computational constraints related to the inversion of large matrices or the requirement of numerous simulations, especially for global datasets (e.g Peylin et al., 2016), and/or the weight of different data streams in the optimisation (e.g. Wutzler and Carvalhais, 2014), may complicate a simultaneous optimisation. On the other hand, in a step-wise assimilation the parameter error covariance matrix has to be propagated at each step, which implies that it can be computed. If the parameter error covariance matrix can

be properly estimated and is propagated between each step, the step-wise approach should be mathematically equal to simultaneous. However, many inversion algorithms (e.g. derivative based methods that use the gradient of the cost function to find its minimum) require assumptions of model (quasi-) linearity and Gaussian parameter and observation error distributions (Tarantola, 1987, p195)."

We have also changed this sentence to explain more what we mean: "If these assumptions are violated, or the error distributions are poorly defined, it is likely that the step-wise will not be equal to the simultaneous, and that information will be lost at each step.", to: "If these assumptions are violated, or the error distributions are poorly defined, it is likely that the step-wise will not be equal to the simultaneous, because information will be lost at each step due to an incorrect calculation of the posterior error covariance matrix at the end of the first step.".

»

2) On page 5 I felt a bit more information was required about the model. How is the value of the functions F(t) being evaluated (possibly I have just misunderstood what is going on - so maybe just some clarification is needed).

» RESPONSE

Indeed we have not described how the function F(t) is calculated at all! Thank you for pointing this out. We have added the following sentence in: "The F(t) forcing term is a random function of time ("log-Markovian" random process) representing the effect of fluctuating light and water availability due to climate on the NPP (Raupach, 2007 – Section 5.3)." Also, following some of the comments from reviewer 1, we have added in further clarifications in this section, including for example the model time step in this sentence: "The first term on the right-hand side of Eq. (1) corresponds to the Net Primary Production (NPP) i.e. the carbon input to the system as a function of time, represented by F(t), weighted by factors (the two fractions in parentheses) that account for the size of both pools, in order to introduce a limitation on NPP.", and "It is based on

two equations that describe the temporal evolution (at a daily time step) of two living biomass (carbon) stores, s1 and s2, and the biomass fluxes between these two stores".

»

3) Page 23, line 4, I am not sure what is meant by orthogonal here. Given that S1 and S2 are interdependent on each other in the quasi-linear model the observations of them (assuming the model is correct, which it is in these synthetic experiments) cannot be not orthogonal. Perhaps the word "additional" would be better used here? Either that or I think the choice of "orthogonal" needs to be justified.

» We agree with the reviewer, this was a lax use of the word in this context. We have changed it to "additional".

Typographic and small errors: P03L10: step -> steps P03L19: one -> only P12L11: uniform -> constant (?) P13L11: than -> as P15L4-L11: this sentence needs to be broken up for clarity. P29L05: 2013.). -> 2013).

» RESPONSE

Thank you for these corrections, we have changed them all. Also we have changed P15 L4-11 from: "Most step-wise test cases (particularly 2b-d) do not result in the same parameter values as the simultaneous test case 3a in which all the observations are included (Fig. 4a), highlighting that strong non-linearity in the model sensitivity to parameters together with the use of an algorithm that is only adapted to weakly non-linear problems, as well as the assumption of linearity in calculating the posterior error covariance matrix at the minimum of the cost function, can result in differences between a step-wise and simultaneous approach in multiple – data stream assimilation (see Section 1)." To: "Most step-wise test cases (particularly 2b-d) do not result in the same parameter values as the simultaneous test case 3a in which all the observations are included (Fig. 4a). This highlights that strong non-linearity in the model sensitivity to parameters, together with the use of an algorithm that is only adapted to weakly

Interactive
comment

non-linear problems, can result in differences between a step-wise and simultaneous approach in multiple – data stream assimilation (see Section 1)."

»

F2a: y-label should read "posterior" instead of "post"? F2b: y label should contain "%". F3caption: Equation should be 1-(RMSE_post/RMSE_prior)x100 F4b: as F2b

» Changed, thank you.

References: Trudinger, Cathy M., et al. "OptIC project: An intercomparison of optimization tech- niques for parameter estimation in terrestrial biogeochemical models." Journal of Geo- physical Research: Biogeosciences 112.G2 (2007). Fox, Andrew, et al. "The REFLEX project: comparing different algorithms and imple- mentations for the inversion of a terrestrial ecosystem model against eddy covariance data." Agricultural and Forest Meteorology 149.10 (2009): 1597-1615. Interactive comment on Geosci. Model Dev. Discuss., doi:10.5194/gmd-2016-25, 2016.
* * *

[revised manuscript text omitted]

---

## Author Comment (AC3) · 25 Aug 2016

UPDATE (24/08/2016) to: Interactive comment on "Consistent assimilation of multiple data streams in a carbon cycle data assimilation system" by Natasha MacBean et al. Anonymous Referee #2

In the first response to reviewer #2 (see above) we responded to the following comment by saying we agreed but needed more time to complete the experiments, and had agreed this with the Editor. This response therefore concerns our update to the point made below, and our "UPDATED RESPONSE" appears after the original response at the end of the document.

I have the following major recommendations to make the manuscript publishable: 1) The experiments performed with the model need to be broader. There are several issues brought up later in the manuscript which could be easily examined. For example some simple experiments looking at populating the off-diagonal elements of the R matrix to set correlation between observations of S1 and S2 would seem to be an easy thing to do. I would be happy to see any sensible additional experiment though.

» RESPONSE

We agree that we could, or should, have added more experiments. Indeed we thought of such experiments from the outset of this work but ended up not including such experiments for fear the paper was too long or the message too complex. We agree that the most obvious, and hopefully most informative, experiment would be one investigating the impact of having correlated observations and populating (or not) the off-diagonal elements of R. We considered other additional tests such as the impact of non Gaussian errors (although we have effectively done this by including an unccounted for bias as described in Section 3.2.2), and we considered examining temporal autocorrelation, but as we want to focus on the multiple data stream aspects we have just considered the correlation between the two data streams. We have implemented this test, but the results we obtained were not what we expected (little impact). As we think this is the most useful extra experiment to include, we have asked the editors for more time (from the 24th June) to investigate this issue and we will provide a further update to this response within the next month. However we will upload the response to the rest of the comments now so the reviewer has more time to look and reply should they wish. The results from these experiments will be presented in a separate section at the end of the experimental section (now Section 3).

»

» UPDATED RESPONSE

We have implemented the experiment the reviewer suggested, that is to test the impact

of correlation between observation errors of the two data streams. We implemented a correlation between the observation errors for each time step of the model following the method of Trudinger et a.l (2007). We could have examined a temporal correlation as well, but as we want to focus on aspects related to multiple data stream assimilation we chose to only look at the cross-correlation between the data streams. We then tested the impact of both accounting for these covariance (correlation) terms in the prior covariance matrix, and ignoring them (i.e. not included them in R). We performed these tests using simultaneous case for both models.

To describe this additional experiment we have changed the text of the manuscript in the following sections:

• Abstract We have added the following sentence: "In addition, we perform a preliminary investigation into the impact of correlated errors between two data streams for two cases, both when the correlated observation errors are included in the prior observation error covariance matrix, and when the correlated errors are ignored."

• Introduction to the experimental results section (now Section 3) We have added this sentence to the introduction to the experimental results section, which itself is an addition to the original submission. The following sentence is an addition to the initial response to the reviewers posted at the end of June 2016.

"In addition to the above three challenges we have performed a preliminary investigation into the impact of correlated errors between the two data streams, which is a topic that has not yet been studied in the context of carbon cycle models"

• Methods section 3.1.6 ("Experiments" – note previously Section 2.1.6 in the original submission) We have added the following paragraph to the end of this section: "For all the above tests wee assumed independence (i.e. uncorrelated errors) for both the parameters and observation covariance matrices, thus the R and B matrices were diagonal. In a final test we performed a simultaneous optimisation to examine the impact of having correlated errors between the s1 and s2 observations. Thus the random

Gaussian noise added to s1 for each time step was correlated to the noise added to s2. The correlated observation errors were generated following the method used by Trudinger et al. (2007 – paragraph 22). The added noise was time invariant, i.e. there was no correlation between one time step and the next as we were specifically looking at correlations between the observations. We tested both accounting for the correlated errors by populating the corresponding off-diagonal elements of the R (observation error covariance) matrix, and ignoring the correlated errors by keeping R diagonal. The reason for performing both tests was to demonstrate the possible real world scenario where correlated observation errors exist, but this information is not included in the optimisation due to a lack of knowledge as to how to characterise the errors. For both tests we performed optimisations using a combination of different of observation error and correlation magnitudes (observations errors between 0.05 and 20 in 9 uneven intervals, and observation correlations between -0.9 and 0.9 with an interval of 0.4). As in the above experiments, twenty random first guesses in the parameter space were used and 15 iterations of the inversion algorithm were performed."

• Finally we have added a whole section to describe the results of this additional experiment – now Section 3.2.5. We will not repeat the text here as it is a clear new standalone section.

We initially found that the model set-up we had used for the set of experiments included in the original submission did not result in any difference when we included the off-diagonal (covariance) terms (accounting for correlation in the observation errors) in the observation covariance matrix (R) compared to when we did not include the off-diagonal terms. This is because the observation errors were small enough to accurately find the minimum of the cost function and the true value of the parameters, and therefore accounting for the correlation in the observation errors had no discernible effect. This was true for any magnitude of observation correlation (postive and negative). We hypothesised that accounting for observation covariance terms (or not) would be an issue if the observation errors were larger (note that larger observation error can be

considered a proxy for anything that would result in lower information in the assimilation system). Therefore we then implemented a test with a range of observation errors and observtion cross-correlation. Indeed above a certain observtion error we did then see a difference between accounting for the off-diagonal terms in R matrix.

These results are described in Section 3.2.5 (entitled "Impact of accounting for correlated observation errors in the prior observation error covariance matrix") and summarised in plots in Figure 7. We highlighted the key finding that at low observation error there is not a discernible difference if you do or do not account for correlated observation errors; however, at higher observtion error (or when the information content of the observations is reduced by another means) it does become important to accurately characterise the correlated errors. We feel this is an important point to make as correlations between observations are largely ignored by the modeling community in parameter optimisation studies, in part because we do not yet have an idea how to characterise the correlations between observations. We have also made the further point, relevant to Section 3.2.3, that accounting for correlation between observations is not possible when performing a step-wise assimilation.

âǎć Perspectives and advice section 4: We updated one bullet point in the advice section from the previous reviewer responses about correlation between observation errors: "Devote time to carefully characterising the parameter and observation error covariance matrices, including their correlations (Raupach et al., 2005), although we appreciate this is not an easy task (but see Kuppel et al., 2013 for practical solutions). In the context of multiple data stream assimilation, this should include the correlation between different data streams, particularly with higher observational uncertainty, though note that this is not possible in a step-wise assimilation."

âǎć Conclusions: Finally we have the following sentence to the Conclusions: "We further note that the consequence of not accounting for cross-correlation between data streams in the prior error covariance matrix becomes more critical with higher observation uncertainty."

Having made all these changes, we also wish to highlight to the reviewer that these experiments have taken some time, in part because this is a new topic that has not yet been fully investigated in any multiple data stream assimilation associated with terrestrial carbon models. As such although we knew what to expect in theory, the detail of results we have obtained beyond the "key finding" discussed above, have puzzled us slightly in that the pattern does not always correspond to our hypotheses. We have made tentative suggestions in the text as to why this is the case, related to non-linearity in the models resulting in inaccurate calculation of the posterior error covariance matrix (as well as higher observation error). We thus feel this topic merits further investigation, a point we have also made in the text. We ourselves plan to continue this investigation topic by starting from scratch and laying out fully our theoretical understanding from a mathematical standpoint using linear model equations. However for this work, given that it is a big topic that may merit a whole study in itself, and given this was suggested as an additional test and we feel we have at least gained one key insight, we hope that the reviewer feels it is a useful addition to this paper. Therefore we are submitting the results of this experiment as they stand for now, despite the fact we would like to (and will) investigate further. We hope that the reviewer now thinks the experimental section is broad enough. Indeed we have tried to further clarify the point of all the experiments (in three main "challenges") by linking them more to the issues related to multiple data stream assimilation in the literature review section (as detailed in the additional response to the reviewer).